# Temperature sensitivity of bat antibodies links metabolic state of bats with antigen-recognition diversity

Nia Toshkova [1,2,7], Violeta Zhelyzkova [1,3,7], Alejandra Reyes-Ruiz [3], Eline Haerens[3], Marina de Castro Deus[3], Robin V. Lacombe[3], Maxime Lecerf [3], Gaelle Gonzalez[4], Nolwenn Jouvenet [5], Cyril Planchais [6] & Jordan D. Dimitrov [3] ✉

The bat immune system features multiple unique properties such as dampened inflammatory responses and increased tissue protection, explaining their long lifespan and tolerance to viral infections. Here, we demonstrated that body temperature fluctuations corresponding to different physiological states in bats exert a large impact on their antibody repertoires. At elevated temperatures typical for flight, IgG from the bat species *Myotis myotis* and *Nyctalus noctula* show elevated antigen binding strength and diversity, recognizing both pathogen-derived antigens and autoantigens. The opposite is observed at temperatures reflecting inactive physiological states. IgG antibodies of human and other mammals, or antibodies of birds do not appear to behave in a similar way. Importantly, diversification of bat antibody specificities results in preferential recognition of damaged endothelial and epithelial cells, indicating an anti-inflammatory function. The temperature-sensitivity of bat antibodies is mediated by the variable regions of immunoglobulin molecules. Additionally, we uncover specific molecular features of bat IgG, such as low thermodynamic stability and implication of hydrophobic interactions in antigen binding as well as high prevalence of polyreactivity. Overall, our results extend the understanding of bat tolerance to disease and inflammation and highlight the link between metabolism and immunity.

Bats (order *Chiroptera*) represent the second largest order of mammals, after rodents, numbering more than 1400 species. They are the only mammals capable of powered flight and possess several exceptional physiological characteristics such as long lifespan, low incidence of age-related diseases, and elevated tolerance to viral infections[1]. Different bat species have been reported to harbor a particularly high diversity of viruses[2–6], some of which can cause severe diseases in humans[7]. Yet, they rarely succumb to immunopathology, thanks to some unique evolutionary adaptations of essential immune pathways[1,7–9]. For instance, bats feature a particular genetic and functional organization of the interferon (IFN) system that can manifest in constitutive tissue expression of IFNs, differential expression of IFN-stimulated genes, and altered kinetics of IFN responses[10,11]. Additionally, in contrast to primates and rodents, many bat species

[1]National Museum of Natural History, Bulgarian Academy of Sciences, Sofia, Bulgaria. [2]Institute of Biodiversity and Ecosystem Research, Bulgarian Academy of Sciences, Sofia, Bulgaria. [3]Centre de Recherche des Cordeliers, INSERM, CNRS, Sorbonne Université, Université Paris Cité, Paris, France. [4]ANSES, INRAE, Ecole Nationale Vétérinaire d'Alfort, UMR Virologie, Laboratoire de Santé Animale, Maisons-Alfort, France. [5]Institut Pasteur, Université de Paris Cité, CNRS UMR3569, Virus Sensing and Signaling Unit, Paris, France. [6]Humoral Immunology Unit, Institut Pasteur, INSERM U1222, Université Paris Cité, Paris, France. [7]These authors contributed equally: Nia Toshkova, Violeta Zhelyzkova. ✉e-mail: jordan.dimitrov@sorbonne-universite.fr

 1

demonstrate impairments in the detection and response to damage-associate molecular patterns (DAMPs), such as cytosolic DNA[12–15].

In addition to the genetic features of their immune components, physiological peculiarities of bats may also play a significant role in tolerance to viral infections. In particular, body temperature in bats can significantly fluctuate on a seasonal and quotidian basis, especially in temperate regions. During the metabolically demanding flight, it is usually around 40 °C, reaching up to 42 °C in some species[16], however, it can drop to ≈20 °C during shallow torpor[17] and even below <10 °C during hibernation[18,19]. Consequently, both high and low body temperatures have been proposed as mechanisms to control infection by providing suboptimal conditions for pathogen replication. In detail, the rise in temperature during flight creates a similar physiological condition as the febrile response in other mammals (so called "flight-as-fever" hypothesis), which is recognized as an evolutionary ancient immune defense mechanism[16,20]. On the other hand, even short torpor bouts have been shown to reduce viral growth rate sufficiently to control infection[17]. Yet, how temperature fluctuations affect the innate and adaptive immune compartments of bats and how this can contribute to increased pathogen tolerance is barely known.

In the present work, we aim to fill this gap by rationally planning experiments based on the specific life history traits and physiological characteristics of bats. In particular, we concentrate on antibodies as the major component of the humoral immune response with central roles in adaptive immune defense against pathogens and maintenance of immune homeostasis[21]. Our data reveal an unconventional mechanism for diversification of immune specificity of IgG antibodies driven by temperature in two bat species, the Greater moused-eared bat (*Myotis myotis*, Borkhausen, 1797) and the Common noctule (*Nyctalus noctula*, Schreber, 1774). Additionally, we describe physicochemical features of bat antibodies that might explain temperature sensitivity. Finally, we identify potential immune defense, homeostatic, and anti-inflammatory functions of this phenomenon that contribute to better understand the mechanisms of the elevated disease tolerance in bats.

## Results

### Temperature modulates the antigen-binding function of bat IgG antibodies

As a consequence of their specific diurnal and seasonal cycles, bats experience large fluctuations in body temperature, and so does their immune system[9,16,18,19,22–26]. This might be particularly relevant to antibodies as they rely on molecular recognition to provide antigen specificity, a process that is strongly affected by a variety of physicochemical factors. Thus, we investigated antigen-binding properties of bat antibodies at three different temperatures: 4 °C, 22 °C and 42 °C, reflecting physiological states of hibernation, shallow torpor, and active flight, respectively. To this end, we purified total IgG from sera of two bat species, *M. myotis* and *N. noctula*, as well as from healthy humans and 4 other mammal species. Next, we analyzed antibody binding as a function of temperature to panels of unrelated mammalian and pathogen-derived proteins and LPS using high-throughput ELISA (Fig. 1a and Supplementary Figs. 1 and 2). The results revealed only negligible reactivity of polyclonal IgG from both bat species to studied antigens at 4 °C and 22 °C. However, the same IgG samples demonstrated significant broadening of their antigen recognition potential at 42 °C (Fig. 1a and Supplementary Figs. 1 and 2). At high temperature, bat antibodies augmented their reactivity to almost all antigens in the panel, yet four proteins were identified as preferentially recognized outliers: NS1 protein of tick-borne encephalitis virus (TBEV), rabies virus glycoprotein, measles virus hemagglutinin, and thyroglobulin (Fig. 1a and Supplementary Fig. 1b). In contrast, immune specificity of human IgG, either pooled or from individual donors, was not significantly affected by temperature (Fig. 1a and Supplementary Fig. 1a), whereby it

demonstrated significant (and independent from temperature within the range of 4–42 °C) binding to LysM (*E. faecalis*), LPS (*E. coli*), SARS-CoV-2 Spike (S) protein, and influenza virus haemagglutinin (Fig. 1a and Supplementary Fig. 1a and Fig. 2). Notably, at 42 °C, human IgG demonstrated a tendency for reduced binding to LPS. Similarly, temperature did not affect (or affected only negligibly) the binding of IgG purified from sera of other mammalian species−mouse, rabbit, goat, and cattle (Supplementary Fig. 2), or the binding of IgY from four bird species (Supplementary Fig. 3). Notably, birds, much like bats have been reported to increase their metabolic rates and heat production during flight[27].

Next, we applied surface plasmon resonance to quantify the effect of temperature on antigen-binding properties of bat antibodies. We assessed antibody binding to the two antigens recognized with the highest intensity in the previous ELISA experiment: NS1 protein of TBEV and rabies virus glycoprotein (Fig. 1b). The real-time kinetic analysis demonstrated that binding of IgG from *M. myotis* to both proteins progressively increased as a function of temperature. At temperatures within the range of 4–16 °C, binding was barely detectable even at the highest antibody concentrations. In contrast, at the highest temperature achievable with the instrument (40 °C), interaction of bat IgG was considerable and subjectable to global kinetic analysis (Supplementary Fig. 4). These showed that bat IgG bound to NS1 protein and rabies virus glycoprotein with physiologically relevant values of apparent binding affinity ($K_D$ values of 26 and 189 nM, respectively; Fig. 1b and Supplementary Fig. 4).

To understand whether the reactivity of bat antibodies is still affected by temperature in a complex biological system, we also performed ELISA experiments using whole sera (Fig. 1c). Again, binding of bat IgG to NS1 of TBEV and rabies virus glycoprotein profoundly increased with temperature, in contrast to binding of human IgG to SARS-CoV-2 S protein and influenza virus HA (Fig. 1a). These results were in complete agreement with our previous findings obtained with purified IgG. However, we noted that the reactivity of bat antibodies to the blocking agent was elevated in complete sera as compared to purified IgG (Supplementary Fig. 5). This could be explained by the possible binding of immunoglobulin classes other than IgG present in complete sera.

To further confirm that temperature sensitivity of bat antibodies is a species-specific and not an individual trait, we used 12 additional serum pools from *M. myotis* to test IgG binding to NS1 of TBEV. As expected, at temperature typical for high metabolic rate (42 °C), there was a marked increase in IgG binding to the antigen (Supplementary Fig. 6a). Importantly, the phenomenon was observed in sera collected both before and after hibernation (Supplementary Fig. 6b), a period associated with significant suppression of immune functions and a decrease in antibody production[28].

Finally, using immunofluorescence, we elucidated the reactivity of bat IgG to HEp-2 cells, which are human cells traditionally used to detect the presence of autoantibodies. Our results indicated only negligible binding of both human and bat (*M. myotis*) IgG to HEp-2 at 22 °C, with bat antibodies demonstrating higher reactivity (Fig. 1d,e). However, at 42 °C, bat antibodies markedly increased their binding to different cellular structures, both cytosolic and nuclear, a typical characteristic of autoantibodies (Fig. 1d,e); this was not observed with human IgG. In addition, elevated temperature significantly increased the reactivity to HEp-2 cells of IgG purified from three other bat species: *Nyctalus noctula, Myotis blythii* (Lasser mouse-eared bat), and *Myotis capaccinii* (Long-fingered bat) (Supplementary Fig. 7).

Collectively, these results show that bat IgG (purified or in whole sera) considerably increases its antigen-binding reactivity towards pathogen-derived and mammalian antigens at temperatures typical for high metabolic rate (observed during flight). Conversely, a rise in temperature does not change the reactivity of antibodies from human, other mammals, as well as birds.

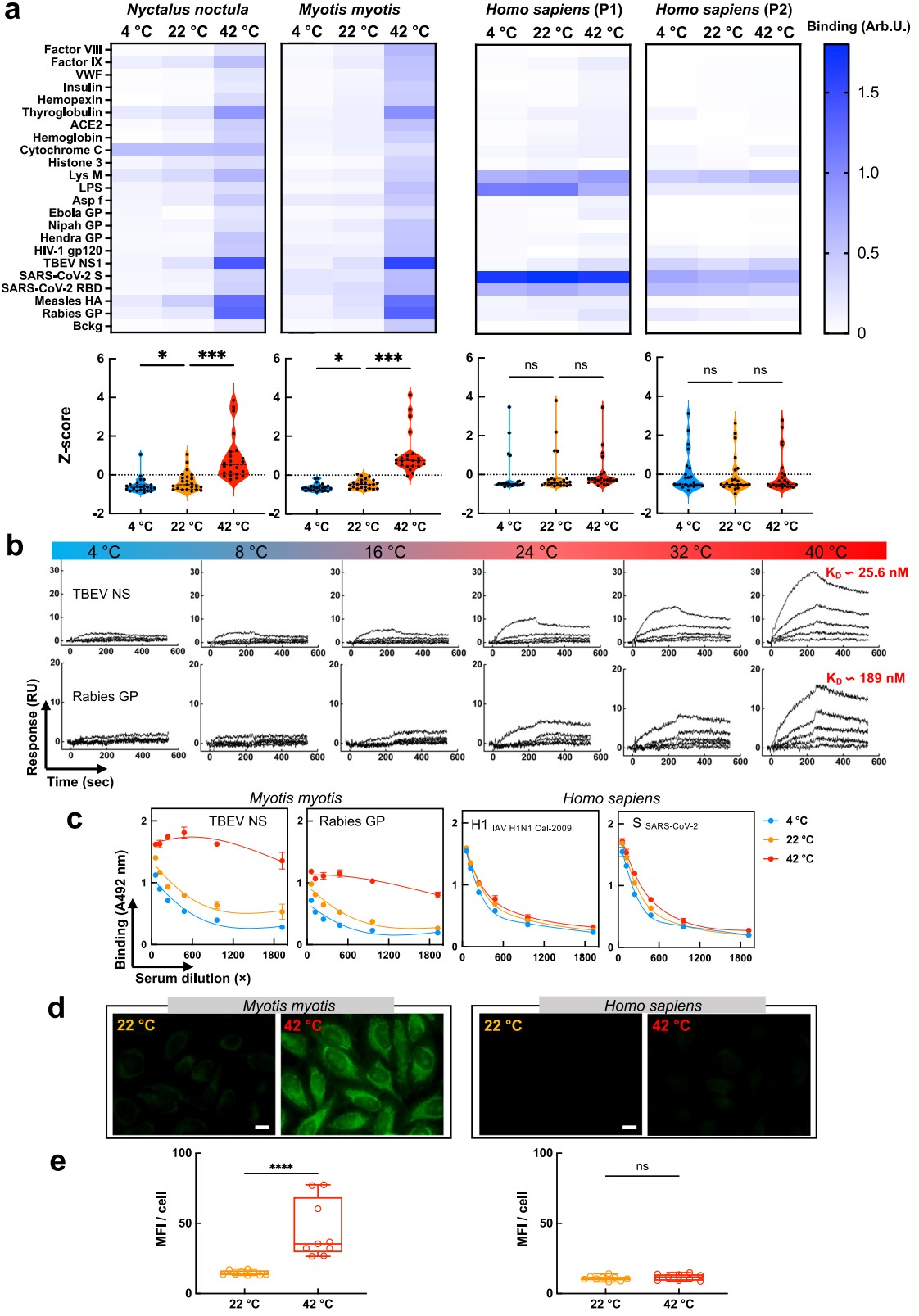

## Temperature broadens diversity of bat antibody-antigen specificities

In order to obtain more integrative insights about the effect of temperature on bat (and human) antibody repertoires, we applied protein array technology. Human proteome arrays HuProt™ display >17,000 different proteins, thus allowing the study of immune reactivity to a huge diversity of targets with different sequences and structural features. In accordance with the previous results (Fig. 1), our pool of purified IgG from *M. myotis* manifested only negligible binding to protein targets at 22 °C. However, when the same pool was probed at 42 °C, we observed a marked broadening in the repertoire of recognized proteins (Fig. 2a). At elevated temperatures, bat IgG recognized a significant fraction of tested proteins (>2400 or ≈14% of all proteins) with a high binding intensity (MFI > 1000), whereby only 19 proteins

**Fig. 1 | Temperature modulates antigen binding properties of bat IgG. a** Heat map depicting binding intensity of IgG purified from pooled sera of *N. noctula*, *M. myotis*, and healthy human (two different pools) to a panel of antigens. The IgG were purified from sera pools with 4–5 individuals. Reactivity of bat and human IgG was evaluated at 4 °C, 22 °C and 42 °C by ELISA. Binding of IgG to the blocking agent (Bckg) is also shown. Binding intensity against each target represents average optical density (*n* = 2 for *M. Myotis* and human; and n = 2 and *n* = 1 for *N. noctula*) after subtraction of the background binding (measured after incubation of the detection reagents with the respective antigens in the absence of antibodies). Violin plots corresponding to each heat map show the Z-score values (*n* = 23). The Z-score for each target antigen and backround is presented as an individual circle. *P* values were determined using one-way analysis of variance (ANOVA) with Friedman multiple comparison test, *\**p* < 0.0153, \*\*\**p* < 0.0007, n.s., not significant. **b** Real-time interaction profiles showing binding of *M. myotis* IgG to Rabies virus glyco-protein and TBEV NS1 immobilized on a sensor chip as a function of temperature. Antibodies were injected at serial dilutions at a concentration range 670–41.875 nM. **c** Effect of temperature on antigen binding by bat and human IgG in whole sera. Pooled bat and human sera were serially diluted from 60 × to 1920 × and incubated with immobilized antigens at 4 °C, 22 °C and 42 °C. Each data point represents average IgG binding intensity ± SD from *n* = 3 technical replicates. **d** Immunofluorescence analyses of bat and human IgG binding to HEp-2 cells at 22 °C and 42 °C. The images are representative examples from *n* = 9 fluorescence acquisitions from two independent experiments (magnification × 63). The white bar corresponds to 10 µm. **e** Quantification of fluorescence intensity from immunofluorescence analyses of binding of bat and human IgG to Hep-2 cells. The graphs show the mean fluorescence intensity of n = 9 images, for each condition, acquired in two independent experiments. Whiskers of the box plots depict minimal and maximal values of MFI, the line represent the median value, the bonds of box correspond to interquartile range. Statistical analyses were performed by a non-parametric t-test, Mann-Whitney two-tailed test, \*\*\*\**p* < 0.0001. Source data are provided as a Source Data file.

were recognized with a high binding intensity at 22 °C. In contrast, temperature affected binding diversity of human IgG to a considerably lower extent (Fig. 2a). At 42 °C, human antibodies recognized 134 proteins with a high binding intensity—a more limited increase in antigen-binding diversity as compared to 22 °C where human antibodies interacted strongly with 8 proteins (Fig. 2a). The considerable difference in the diversity of bat and human immune specificities is particularly obvious when plotted together (Fig. 2b): even though at 22 °C bat antibodies displayed slightly elevated immune reactivity as compared to human, at 42 °C a considerable asymmetry in the protein binding landscape was evident (Fig. 2b). Finally, the plots of Z-scores indicated that at high temperature, a larger set of proteins was recognized with significantly higher than average binding intensity by bat IgG as compared to human IgG (Fig. 2c). Similar results were obtained from *N. noctula* where antigen binding diversity of pooled IgG also considerably increased at 42 °C as compared to 22 °C (Supplementary Fig. 8).

These data indicate that bat antibodies can significantly broaden their antigen binding diversity at temperatures typical for active metabolic state. On the contrary, antigen binding diversity of human antibody repertoire is much less affected by temperature variation.

### Temperature differentially affects Fab- and Fc-dependent interactions of bat antibodies

Next, we performed experiments to rule out possible non-specific effects of temperature on bat IgG molecules. First, our aim was to establish if other bat proteins change their specificity as a function of temperature. Thus, binding of serum albumin purified from *M. myotis* to the proteins available in a crude extract of *M. myotis* epithelial cells was not affected by an increase in temperature, contrary to the behavior of IgG purified from the same serum pool (Supplementary Fig. 9). With this experiment we showed that the broadened reactivity as a function of temperature is not typical for all bat proteins.

To demonstrate further that temperature specifically affects the antigen-binding region of bat antibodies, we generated F(ab')₂ fragments from bat IgG by enzymatic digestion with the bacterial protease IdeS (Fig. 3a). As with whole IgG, binding of bat F(ab')₂ to distinct antigens strongly increased as a function of temperature (Fig. 3b).

Next, we focused on the effect of temperature on interactions mediated by the Fc portion of IgG molecules. Comparison of sequences of the constant portions of IgG1 of *M. myotis* and human showed relatively high degree of homology (ca. 67%, Supplementary Fig. 10). It is noteworthy that the identity of this region between *M. myotis* and murine IgG subclasses was lower (ca. 58% for murine IgG2b, Supplementary Fig. 10). The high sequence identity of their constant portions can explain the potential of human Fc-binding proteins, such as neonatal Fc receptor (FcRn), to bind both human and bat IgG with high affinities[29]. This allowed us to compare the effect of temperature

on the Fc-mediated interactions of bat (*M. myotis* and *N. noctula*) and human IgG with distinct Fc-specific proteins (Fig. 3c). Real-time binding profiles and kinetic analyses of interactions with complement protein C1q (Fig. 3d), FcRn receptor (Fig. 3e) and protein G (Fig. 3f) showed that temperature negatively impacts binding of all studied proteins to both human and bat IgG (Fig. 3d, e, f)—an effect opposite to the one observed for the Fab-dependent interactions.

Collectively, the results obtained indicate that the phenomenon of increasing antigen binding diversity in bat IgG at elevated temperatures is specifically associated with the variable regions of antibodies.

### Particular physicochemical qualities of bat antibodies can contribute to their temperature sensitivity

To get an insight into the molecular mechanism of the noncanonical diversification of immune specificities in bat antibodies, we studied their molecular properties. First, we estimated the thermodynamic stability of IgG molecules using a thermal shift assay. The unfolding temperature (or melting point Tm) of *M. myotis* IgG was significantly lower than that of human IgG (Fig. 4a), indicating lower thermodynamic stability. This was also confirmed using another bat species, *Myotis capaccinii*, (Supplementary Fig. 11). It is important to note that this assay measures global stability of IgG and cannot distinguish the difference in stability of different domains or fragments of the molecule.

Additionally, we compared the overall hydrophobicity of IgG from human and *M. myotis* using hydrophobic interaction chromatography. The resulting elution profiles indicated that the overall hydrophobicity of bat and human IgG molecules showed negligible difference (Fig. 4b). This is in accordance with the relatively high sequence similarity of bat and human Fc fragments (Supplementary Fig. 10). The estimation of bulk antibody hydrophobicity, however, cannot provide details about the implications of different types of non-covalent forces during the interaction with an antigen. To this end, we studied the effect of ionic strength on the antibody-antigen binding process (Fig. 4c). The IgG from the two bat species, *M. myotis* and *N. noctula*, showed specific V-shaped responses to variations in NaCl concentration upon binding to distinct antigens (Fig. 4c). This pattern was different from the one observed in human antibodies. Interaction of bat antibodies with antigens was favored in an environment with both low and high ionic strength. This implies antigen binding involving both polar and hydrophobic components (Fig. 4c). Notably, these data also suggest that at physiological concentration of NaCl (0.15 M), the contribution of polar interactions is highly reduced and binding to antigens is primarily driven by hydrophobic forces (Fig. 4c).

To better understand the difference in binding behavior of bat and human antibodies, we analyzed the sequences of human and bat antibody variable regions. As annotated V$_H$ family gene sequences are not yet available for *M. myotis* or any of our study species, we used those from a previous work on the closely related *Myotis lucifugus*[30].

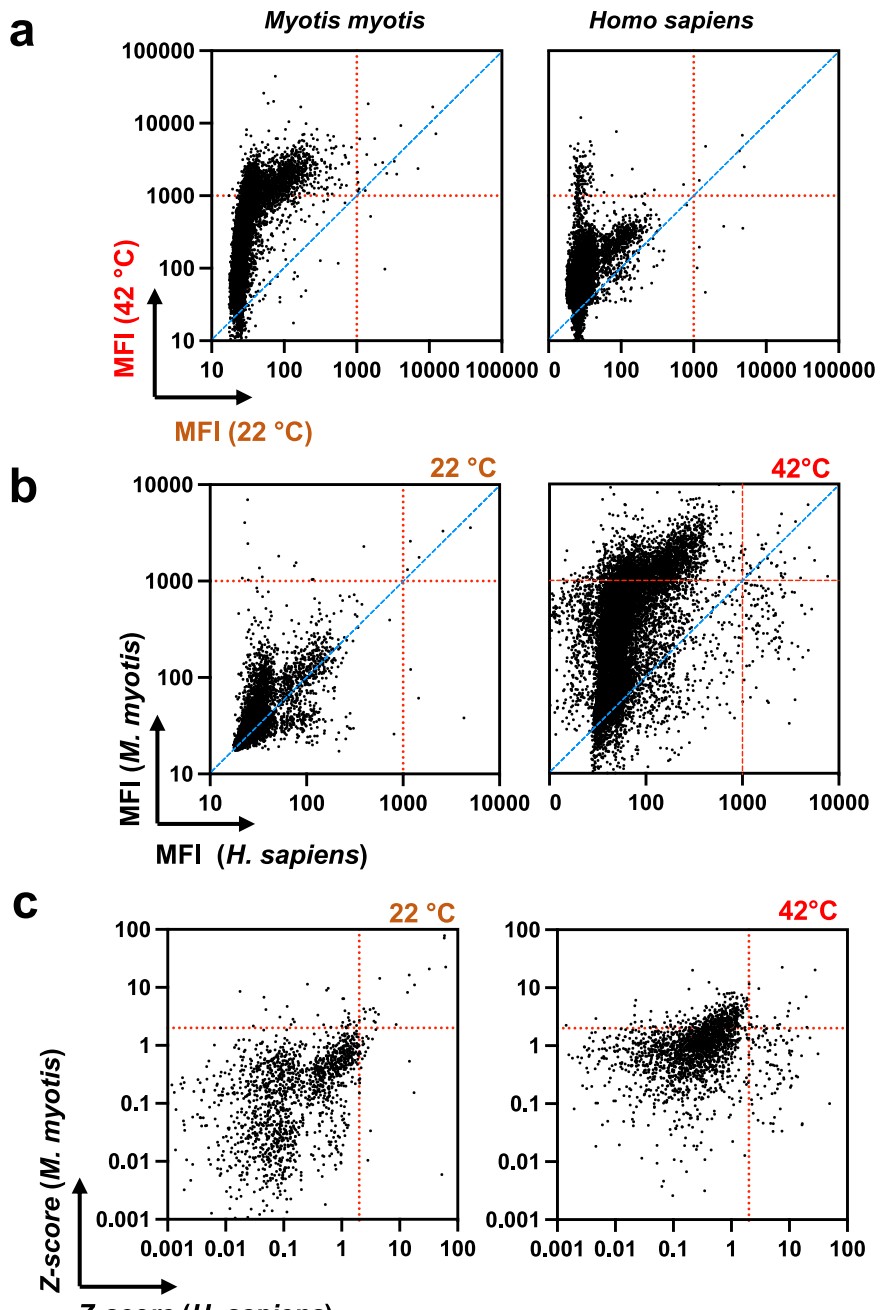

**Fig. 2 | Bat antibodies acquire broad reactivity towards diverse proteins at elevated temperature.** Presented plots depict summarized results from protein microarray analyses of binding of bat (*M. myotis*) and human pooled IgG to >17,000 human proteins. **a** Antibody reactivity at 22 °C was plotted versus reactivity at 42 °C; **b** Reactivities of bat versus human IgG. Each dot in (**a**) and (**b**) represent the average MFI ($n = 2$) signifying IgG binding to a single protein in the array, after subtraction of the background reactivity. Red dashed lines represent the threshold of 1000 MFI, identifying antibody specificities with high binding intensity.

**c** Z-scores of antibody binding reactivities to human proteins. Each dot represents the Z-score characterizing binding reactivity to a single protein. The threshold indicated with red dotted lines corresponds to $Z = 2$ and $p < 0.05$. **a**–**c** Statistical analyses were performed by a nonparametric t-test, Wilcoxon two-tailed test to compare the mean IgG reactivities for different conditions. $p < 0.0001$ values were obtained for all data plots presented in the figure. Source data are provided as a Source Data file.

Sequence comparison of bat ($n = 75$) and human ($n = 56$) $V_H3$ genes (the most frequently used in human) revealed no difference in their overall organization. However, in silico analysis (computing the GRAVY index) showed that heavy chain variable regions of bat antibodies were characterized by significantly higher hydrophobicity (less negative GRAVY index value, Fig. 4d), most evident in framework region 1 and CDR H2 (Fig. 4d).

Taken together, the results underline several physicochemical differences in bat antibodies that are not typically found in their human counterparts. These specific qualities might be associated with untypical binding properties of bat antibodies as a function of temperature.

**Higher prevalence of polyreactive antibodies in bat immune repertoires is associated with temperature sensitivity**
Previous studies on murine monoclonal IgG and human Fab fragments have demonstrated that polyreactive antibodies (or Fab), contrary to monoreactive ones, bind to an increased number of antigens at high

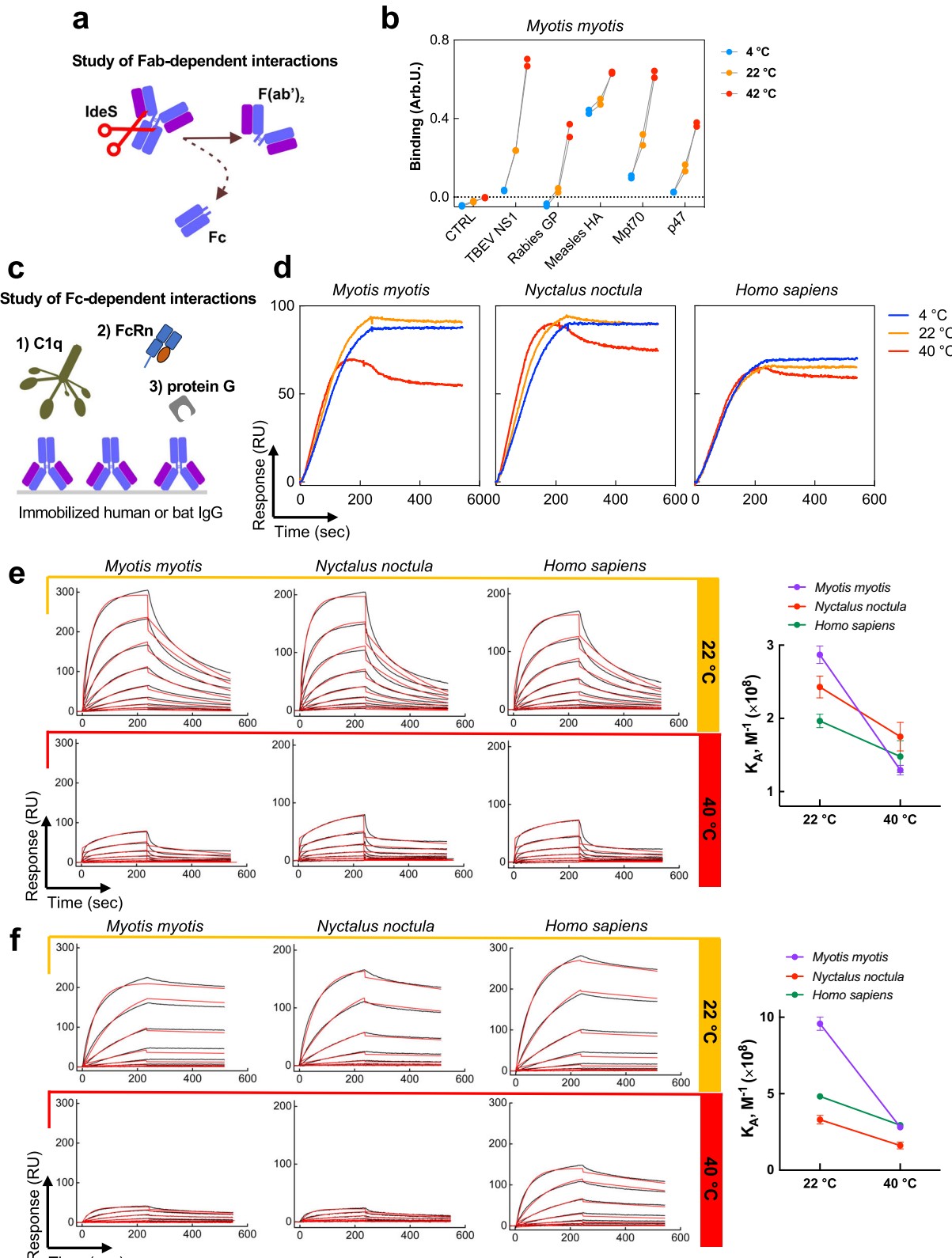

temperatures[31,32]. To contextualize these findings within our study, we analyzed the reactivity of ED38, a widely used human monoclonal polyreactive antibody, as well as of two monoreactive antibodies: VRC01 (specific for gp120 of HIV-1) and S309 (binding to S proteins of both SARS-CoV-1 and SARS-CoV-2). We performed an ELISA at 4 °C, 22 °C and 42 °C using the same panel of antigens that we used for our bat IgG experiments (Fig. 1a). As expected, the polyreactive ED38

significantly increased its binding to most antigens as a function of temperature (Fig. 5a and Supplementary Fig. 12). An opposite effect was even observed in the case of monoreactive IgG1 (VRC01), where binding to gp120 slightly decreased at 42 °C (Fig. 5a). To test whether temperature-sensitivity of bat IgGs was mediated by elevated prevalence of polyreactive antibodies, we applied a molecular probe, dinitrophenol (DNP) conjugated to BSA. This has been proven reliable

**Fig. 3 | Temperature sensitivity of bat IgG is mediated by Fab. a** A schematic representation of the experimental procedure for the generation of F(ab')$_2$ fragments from IgG of *M. myotis*. **b** Reactivity of bat F(ab')$_2$ to a panel of antigens as a function of temperature. Binding intensity values were obtained after incubation of 10 µg/ml (67 nM) of F(ab')$_2$ fragments with a panel of immobilized antigens at 4 °C (blue), 22 °C (orange), and 42 °C (red). Each point corresponds to an individual technical replicate for each condition after subtraction of the background (obtained following incubation of the detection reagents with the respective antigens in the absence of F(ab')$_2$ fragments). Control reactivity of bat F(ab')$_2$ to the surface in the absence of antigens is also shown. **c** A schematic representation of the experimental setting for the evaluation of interactions of C1q, FcRn, and protein G with the Fc-fragment of bat IgG. **d** Real-time interaction profiles of binding of human C1q (0.625 nM) with immobilized IgG from *M. myotis*, *N. noctula* and human. Binding curves obtained at 4 °C (blue), 22 °C (orange), and 40 °C (red) are depicted. Real-time interaction curves were obtained after subtraction of C1q binding to a control surface without immobilized IgG. **e** Real-time interaction profiles showing binding of human recombinant FcRn to IgG from *M. myotis*, *N. noctula* and human immobilized on sensor chip. FcRn was injected at serial dilutions in concentration range 25–0.097 nM (2 × dilution step) in a buffer with pH 5.7. **f** Real-time interaction profiles showing binding of recombinant protein G to IgG from *M. myotis*, *N. noctula*, and human immobilized on the sensor chip. Protein G was injected at serial dilutions in concentration range 100–0.046 nM (3 × dilution step). For (**e**) and (**f**) Measurements were performed at 22 °C and 40 °C. The mean values of equilibrium association constants ($K_A$ shown in plots on the right); ±SD ($n$ = 3) were evaluated by global Langmuir kinetics analyses model from their independent global fit analyses. The real-time binding profiles after subtraction of reactivity to a control surface are shown by black lines. The global kinetic fit is depicted by red lines. For example, representative results from two independent experiments are shown. Source data are provided as a Source Data file.

for the estimation of overall levels of polyreactive antibodies in antibody repertoires[33,34]. Indeed, IgG from *M. myotis* and *N. noctula* demonstrated a considerably higher reactivity (2–3-fold) to immobilized DNP at 22 °C as compared to human IgG (Fig. 5b). The same probe successfully distinguished polyreactive from monoreactive antibodies (Fig. 5c). Similar results were obtained with additional probe for polyreactivity—the hydrophobic hapten protoporphyrin IX (Supplementary Fig. 13), as well as by immunoblot and immunofluorescence analyses (Fig. 5d, e). There, IgG from *M. myotis* and *N. noctula* demonstrated high reactivity to proteins present in a *M. myotis* epithelial cell lysate at 22 °C (MmNep, Fig. 5d). Notably, this binding further increased at lower ionic strength (Fig. 5d)—another behavior typical for polyreactive antibodies[35]. Polyreactivity and autoreactivity of bat IgG were also evident by the presence of substantial binding to autologous epithelial cells (Fig. 5e). Thus, this interaction was detectable even at 22 °C; albeit at significantly lower extent as compared to elevated temperature (Fig. 5e, f), In contrast, human IgG bound only negligibly to human cells at both 22 °C and 42 °C (Fig. 1d, Fig. 6).

Overall, these data suggest that bat antibody repertoires contain considerably higher levels of polyreactive antibodies. This could partly account for the diversification of their antigen-binding reactivity at high temperatures.

**Elevated temperature preferentially triggers the reactivity of bat antibodies to dead cells**

To gain insight into the physiological role of temperature-sensitive IgG, we investigated the capacity of bat and human antibodies to opsonize human endothelial cells. The marked intensification of bat metabolic rate during flight is associated not only with increased body temperature but also with the release of ROS, which can result in cell death and liberation of various endogenous molecules with pro-inflammatory activity (DAMPs)[1]. The temperature-dependent augmentation of antibody reactivity to mammalian proteins may serve as a mechanism for clearing damaged or dead cells, which possess potent pro-inflammatory potential. To test this hypothesis, we modeled induction of cell death by oxidative stress through the exposure of human endothelial cells (HMEC-1) to hydrogen peroxide. The gating strategy for identification of necrotic endothelial cells is presented in Fig. 6a. Labeling cells with annexin V and propidium iodide clearly indicated that exposure to oxidative stress triggered endothelial cell death. To assess the impact of temperature on the reactivity of *M. myotis* and human IgG to cells with different viability, we incubated antibodies at 22 and 42 °C and tested their capacity to opsonize alive and dead cells by flow cytometry (Fig. 6b). Both human and bat IgG exhibited low reactivity to alive HMEC-1 cells, which remained largely unchanged when incubation was performed at high temperature (Fig. 6b). Antibody reactivity to dead cells at 22 °C was also low for both human and bat samples, yet bat antibodies demonstrated a significant increase in binding (Fig. 6c). In contrast, at 42 °C, binding of

bat IgG to dead cells was significantly augmented (Fig. 6b, c), while reactivity of human IgG was not modified by the increase in temperature.

To extend these studies with bat autologous cells, we evaluated the reactivity of *M. myotis* IgG to live and dead epithelial cells from the same species (Fig. 7). Those experiments demonstrated the capacity of bat IgG to bind to live and dead cells at 22 °C (Fig. 7b), which further confirmed the high frequency of polyreactivity and autoreactivity in bat antibody repertoires (Fig. 5). Importantly, at elevated temperature (42 °C), bat antibodies significantly increased their binding to dead but not to alive epithelial cells (Fig. 7b, c). These results are consistent with the data obtained with endothelial cells (Fig. 6).

Together, these data indicate that bat antibodies acquire preferential reactivity to dead endothelial and epithelial cells at elevated temperatures. This likely facilitates the efficient opsonization and rapid clearance of such cells during a metabolically active physiological state, thereby exerting an important anti-inflammatory effect.

## Discussion

The present study draws a direct link between specific life history traits and physiological characteristics of bats and the function of their immune compartments. In particular, we demonstrated that temperature fluctuations, accompanying daily and seasonal cycles of bats, can regulate the extent of antigen binding strength and diversity in their antibody repertoires. By elucidating the reactivity of IgG antibodies from two temperate bat species, the Greater mouse-eared bat (*Myotis myotis*) and the Common noctule (*Nyctalus noctula*), we found that elevated temperatures (40–42 °C), typical for the active metabolic state during flight, significantly broadened their spectrum of recognized antigens. On the other hand, at temperatures representing inactive metabolic states (shallow torpor, 22 °C, or hibernation, 4 °C), diversity of antigen binding specificities of bat antibodies substantially collapsed. The same temperature changes did not modulate binding diversity of human antibodies or antibodies from other mammalian and bird species. Of note, these results are derived from several independent experiments with multiple bat serum pools, collected at different times of the year (including before and after hibernation) from three different locations. This reliably proves that antibody temperature-sensitivity is a stable trait in our study species. Furthermore, the application of highly diverse protein microarrays (containing >17,000 proteins) in the study indicates that the observed phenomenon is not limited to a particular type of antigens and might have a broad physiological relevance. Lastly, obtaining equivalent results from species belonging to two genera with distinct life history traits indicates that antibody temperature-sensitivity might be widely distributed among order Chiroptera.

The study also demonstrated that the sensitivity of bat antibodies to temperature fluctuations is mediated by the antigen-binding fragment of the immunoglobulin molecule but not by regions in its

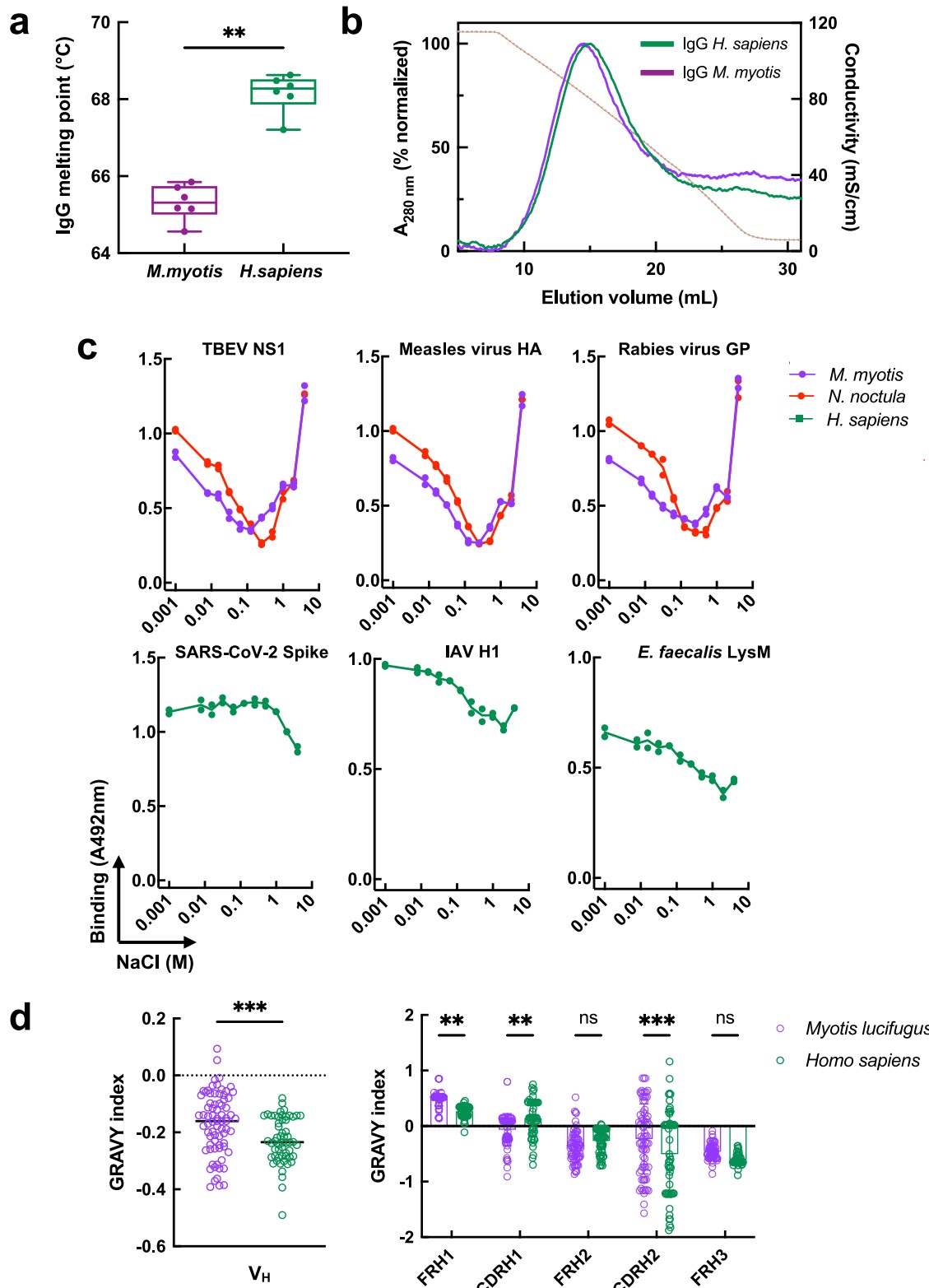

constant portion. Previous research demonstrated that interactions of murine polyreactive antibodies, or human polyreactive Fab fragments, were exclusively augmented at high temperatures in contrast to those of monoreactive antibodies (or Fab fragments)[31,32], an effect that we confirmed here with human monoclonal IgGs. Indeed, our data from different orthogonal approaches indicates that bat IgG repertoires have a higher prevalence of polyreactive antibodies as compared to human repertoires. This likely contributes to temperature-sensitivity

and antibody repertoire diversification in these species. Yet, we do not exclude the implication of other (not mutually exclusive) mechanisms, especially in the cases of antigens recognized with high binding affinities (NS1 of TBEV and rabies virus glycoprotein). Indeed, our biochemical analyses revealed some unusual physicochemical features of bat IgG, such as significantly lower thermodynamic stability and distinctive implication of both polar and hydrophobic forces in the antigen binding process. The thermodynamic stability of proteins is

**Fig. 4 | Bat IgG is characterized by untypical physiochemical characteristics.**
**a** Melting temperatures of bat (*M. myotis*) and human IgG, obtained by thermal-shift assay. Each dot indicates an individual estimation of the melting temperature (Tm, °C). The results of two independent experiments with *n* = 6 measurements of Tm per species are presented together. Whiskers of the box plots depict minimal and maximal values of Tm, the line represent the median value, the bonds of box correspond to interquartile range. Statistical significance was assessed by applying a nonparametric t-test analysis, Mann-Whitney two-tailed test. **$p$ = 0.0022. A comparison of the thermodynamic stability of IgG from another bat species (*M. capaccinii*) with the stability of human IgG is depicted in Supplementary Fig. 11. **b** Overall hydrophobicity of human and bat IgG. Elution profiles of polyclonal human and *M. myotis* IgG (green and purple lines, respectively) obtained by hydrophobic interaction chromatography. Chromatography profiles were obtained by measuring protein absorbance changes at 280 nm. The normalized profiles are presented. The brown dotted line depicts the elution gradient of buffer with a decreasing ionic strength as assessed by changes in conductivity. A representative result from two independent experiments is shown. **c** Ionic strength dependence of antigen binding of bat and human IgG. Polyclonal IgG from *M. myotis* (purple symbols and line), *N. noctula* (red symbols and line), and human (green symbols and line) were diluted to 20 μg/ml in 10 mM HEPES buffer containing increasing concentrations of NaCl: 0, 0.0078, 0.0156, 0.0312, 0.0625, 0.125, 0.25, 0.5, 1, 2, and 4 M and incubated with the indicated proteins immobilized on ELISA plate. Each data point represents absorbance at 492 nm from (*n* = 2) technical replicates. **d** In silico comparison of the hydrophobicity of the variable region of human and *Myotis lucifugus* heavy immunoglobulin chains. The hydrophobicity index was calculated by GRAVY calculator (https://www.gravy-calculator.de) (in bulk or by subregions) using amino acid sequences taken from[30] (*M. lucifugus* *n* = 75) and[69] (human, *n* = 56). The lines and the bar sizes in the figures represent the mean values. Statistical analyses were performed on the obtained GRAVY values by applying a nonparametric t-test analysis, Mann-Whitney two-tailed test. **$p$ = 0.0006 (comparison of whole VH) and Two-way ANOVA, with Šídák's multiple comparisons test (for comparison of different subregions), **$p$ = 0.0018 (FRH1), **$p$ = 0.0066 (CDRH1), ***$p$ = 0.0001 (CDRH2). Source data are provided as a Source Data file.

determined by the complexity of noncovalent interaction networks maintaining the protein fold. Thus, polypeptide chains of less stable proteins are usually less tightly packed and characterized by increased structural dynamics[36,37], which can be further enhanced as a function of temperature. This is probably the case of bat antibodies where higher flexibility of the antigen-binding site probably allows increased binding adaptability, recognition of a large number of antigens[38–40], or targeting epitopes difficult to recognize by conventional antibodies[41]. Another physiochemical quality of bat IgG is their reliance on polar and hydrophobic forces for antigen binding upon variation of ionic strength. Thus, despite the presence of polar groups in antigen-binding sites, at physiologically relevant ionic strength, bat antibody-antigen interaction seems to be mainly driven by hydrophobic forces. This agrees with the sequence analyses of immunoglobulin genes showing that some regions in heavy chain variable domains (framework region 1 and CDRH2) in bats have significantly higher hydrophobicity as compared to their human counterparts. Together, these findings may explain the temperature effects as it was found that molecular interactions that are mainly driven by hydrophobic forces increase their strength as function of temperature[42,43]—a fact in full concordance with our data.

The unconventional mechanism for control of antigen binding diversity, proposed here, could further explain disease tolerance in bats and may also have other important biological repercussions. Immunogenetic studies have demonstrated that the diversity of functional gene segments encoding variable immunoglobulin regions in bats is similar to that of humans and mice[30,44–46]. Thus, the potential repertoire of bat antibody specificities seems sufficiently large to cope with high versatility of pathogens, without the need for any extraordinary diversification strategies. Yet, immune reactivity of bat antibodies that correlates to (and is controlled by) the state of metabolic activity might be biologically meaningful by providing a physiological adaptation for optimizing immune response timing. For instance, reducing antibody reactivity at low temperatures might be an additional strategy to economize energy and resources during phases of inactivity and suppressed metabolism, i.e., daily torpor or seasonal hibernation[47,48]. As pathogen replication is severely inhibited at low temperatures typical for torpor, some immune components may not be necessary to control infection during this period. This is particularly true for adaptive immune responses that are known to be one of the most energy-demanding processes in mammals[49,50]. On the other hand, substantial antibody diversification observed at high temperatures may provide bats with instant protection upon exiting torpor and returning to body temperatures that are favorable for pathogen replication. Such acting at the first line of immune defense is typical for polyreactive antibodies that allow the immune system to rapidly

cover a practically infinite antigenic space, albeit with lower affinity. Interestingly, however, bat IgG also shows a temperature-dependent rise in binding strength to specific antigens as exemplified by our ELISA and real-time kinetic experiments with NS1 protein of TBEV and rabies virus glycoprotein. Both have been previously detected in the bat species used in our study[51–54]. Therefore, the apparent binding affinity of bat IgG to them at high temperature may indicate immune repertoire shaping as a result of coevolution with these or other related viruses.

Another important aspect of bat antibody temperature sensitivity is the potential to control inflammation induced by the accumulation of dead cells and the release of DAMPs during active metabolic states. This is in full concordance with previous research suggesting the evolution of powered flight as a main driver of immune system evolution in bats[9,55,56]. Indeed, flight in bats is associated with a substantial increase in metabolic rate (>30 folds), rise in body temperature, and intensification of oxidative processes that can lead to cell damage and the release of various DAMP molecules such as DNA, misfolded proteins, oxidized phospholipids, ATP, etc[1,22,57,58]. As an adaptation to such metabolic stress, the activity of innate pathways sensing these molecules is reduced in bats, thus protecting them from perpetual inflammation[8,9,55]—a phenomenon that has also been associated with bat tolerance to viral infections[9,59]. Our study provides yet another anti-inflammatory mechanism in bats by showing that bat IgG preferentially binds and opsonizes dead human endothelial or bat epithelial cells while sparing living cells at elevated temperatures. Thus, the increase in antibody reactivity during active metabolic states probably contributes to the clearance of apoptotic cells and damaged macromolecules in vivo. The distinction between healthy and damaged cells by bat antibodies can be based on one or several hallmarks typical for dying eukaryotic cells—changes in membrane phospholipids composition, membrane permeabilization, appearance of cryptic and oxidized epitopes, etc[60–62]. Studies in mouse models and humans have confirmed the important role of natural polyreactive antibodies in the maintenance of immune homeostasis through the clearance of damaged or dead cells[60,63–65]. This is in accordance with our data showing that a considerable fraction of bat antibodies are polyreactive.

Taken together, the findings from this study uncover new features of bat immune system and provide additional explanation of bat tolerance to inflammation and disease. We also depict an alternative mechanism for diversification of immune specificity of antibodies in general and highlight the link between metabolism and immunity. Further elucidating the mechanism of temperature sensitivity of bat antibodies will extend our understanding about evolution of immune system in mammals and can lead to practical discoveries with important repercussions for human health.

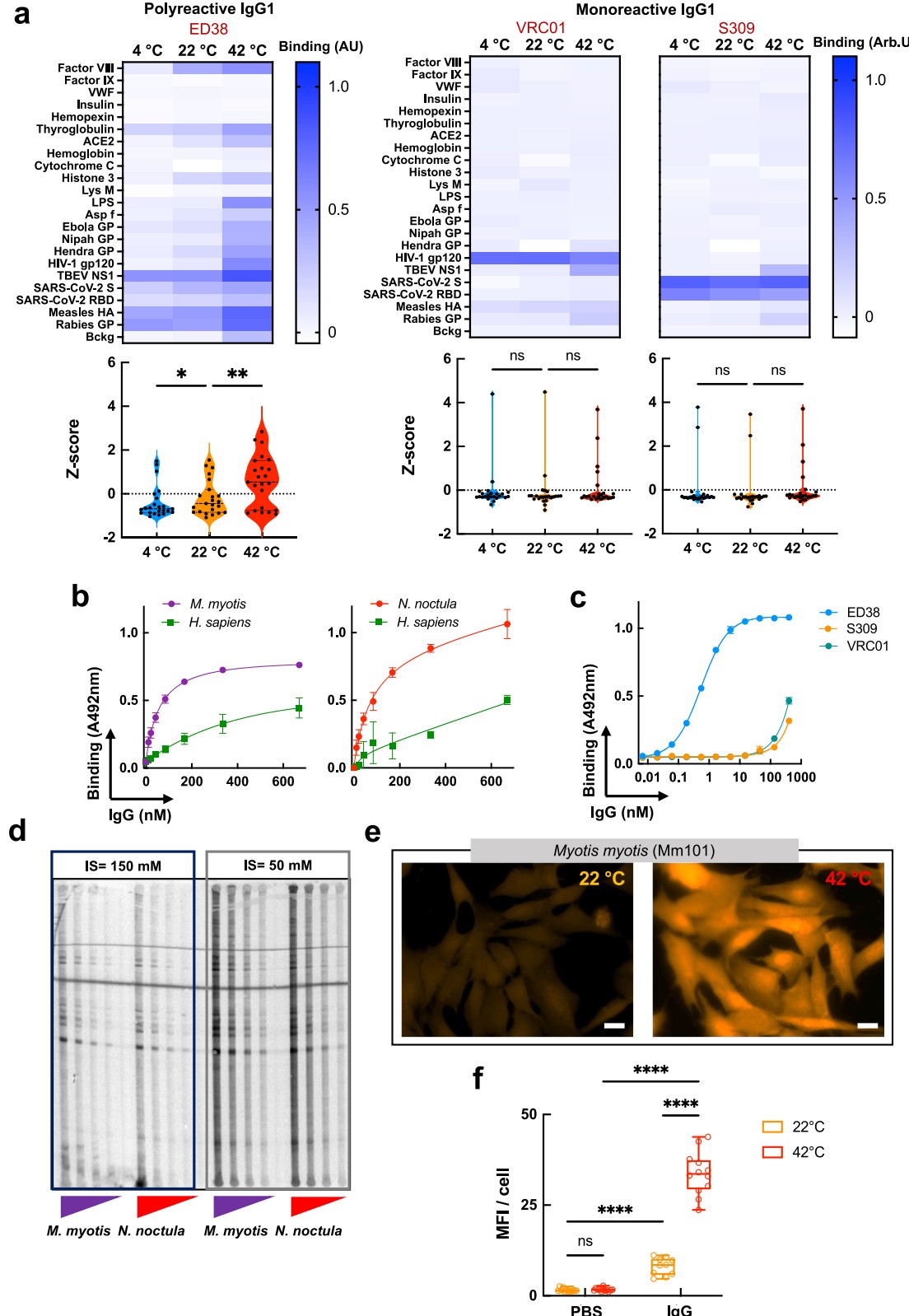

## Methods

### Bat capture and blood sampling

The research was carried out under permit by the Bulgarian Biodiversity Act (No 830/19.09.2020 and No 927/04.04.2022). For this study, we used samples from four bat species from two genera: Greater moused-eared bat (*Myotis myotis*), Lesser mouse-eared bat (*Myotis blythii*), Long-fingered bat (*Myotis capaccinii*), and Common noctule

(*Nyctalus noctula*). Greater moused-eared bats were caught at Ivanova voda cave, situated in Western Rhodopes mountain, Bulgaria (N41.894, E24.880), in April 2021 (*n* = 40), October 2021 (*n* = 45), May 2022 (*n* = 29) and October 2023 (*n* = 20) as well as in Orlova chuka cave, situated in Northern Bulgaria (N 43.5932, E 25.9601), in September 2023 (*n* = 50). Lesser mouse-eared bats were caught in Ivanova voda in October 2023 (*n* = 18) and in Orlova Chuka cave in September 2023

**Fig. 5 | Bat antibody repertoires have higher levels of polyreactive IgG antibodies. a** Heat maps depicting binding intensity of human polyreactive (ED38) and monoreactive (VRC01 and S309) IgG1 to a set of distinct antigens. Reactivity of monoclonal antibodies was evaluated at 4 °C, 22 °C, and 42 °C by ELISA. Binding of IgG to the blocking agent (Bckg) is also shown. Binding intensity against each target represent average optical density (*n* = 2) after subtraction of the background binding measured after incubation of the detection reagents with the respective antigens in the absence of IgG. For the analyses, all antibodies were diluted to 10 µg/ml. Violin plots corresponding to each heat map show the Z-score values of antigen binding (*n* = 23). The Z-score for each target antigen and background is presented as an individual circle. *P* values were determined using one-way analysis of variance (ANOVA) with Friedman multiple comparison test, *\**p* = 0.0153, \*\**p* = 0.0035, n.s., not significant. Binding of increasing concentrations of human and bat (*M. myotis* and *N. noctula*) IgG antibodies (**b**) and human monoclonal IgG1 antibodies (**c**) to immobilized DNP-BSA. Each data point on the graphs represents average optical density (*n* = 3) ± SD. Representative results from two independent experiments are shown. **d** Immunoblot analyses of bat IgG binding to lysate of bat epithelial cells. The figure shows the reactivity of increasing concentrations (2.5–20 µg/ml) of IgG from *M. myotis* and *N. noctula* towards proteins from crude extract of MmNep epithelial cells from *M. myotis*. The antibodies were alternatively diluted in buffer (PBS) with physiological and low ionic strength. A representative example from two independent experiments is shown. **e.** Immunofluorescence analyses of binding of *M. myotis* IgG to MmNep cells at 22 and 42 °C. The images are representative examples from *n* = 12 fluorescence acquisitions from one of two independent experiments (magnification ×63). The white bar corresponds to 10 µm. **f** Quantification of fluorescence intensity from immunofluorescence analyses of bat IgG binding to MmNep cells. The graph shows the mean fluorescence intensity of *n* = 12 images. Whiskers of the box plots depict minimal and maximal values of MFI, the line represent the median value, and the bonds of box correspond to interquartile range. Statistical analyses were performed using Two-way ANOVA Tukey's multiple comparison test, \*\*\*\**p* < 0.0001. Source data are provided as a Source Data file.

---

(*n* = 10). Long-fingered bats were also caught at Ivanova voda in April 2021 (*n* = 3), May 2022 (*n* = 20), and October 2023 (*n* = 12). Noctule bats were caught at Devetashkata cave, situated in North Central Bulgaria (N43.233, E24.885), in June 2022 (*n* = 18) and August 2023 (*n* = 71). All bats were caught at cave entrances at sunset using a harptrap (for Ivanova voda and Orlova chuka) or mistnets (for Devetashkata cave), then they were kept in individual cotton bags until processing. Their sex, size, weight, and age group (adult/juvenile) were noted.

Blood samples were taken from the uropatagium vein by puncturing it with a sterile disposable insulin needle and collecting the few drops of leaked blood (most often below 100 µL) using a sterile disposable hematocrit capillary or a pipette with sterile disposable tips. Blood sampling of this kind is not supposed to affect the survival rates of bats[66]. Blood serum was immediately separated from plasma by centrifugation for 15 to 30 mins at 1000 × *g*. Separated sera were then frozen at −20 °C. Each sampled bat was given water and mealworms (*Tenebrio*) to compensate for blood loss when appropriate. We minimized handling time to reduce stress on the animals. All researchers used personal protective equipment (masks, gloves) and non-disposable instruments and working surfaces were disinfected between each animal. To obtain enough material for the experiments, usually we pooled sera from 3 to 10 individuals directly in the field or in the lab.

### Samples from human and other species
In the study, we used fresh AB serum samples from ten anonymised healthy human donors, obtained under the ethical convention between INSERM with Etablissement Français du Sang #18/EFS/033. Pooled sera from different mammals: mouse (*Mus musculus*, #NS03L), rabbit (*Oryctolagus cuniculus*, #R4505), goat (*Capra hircus*, #G9023), and cattle (*Bos taurus*, #S1507) were obtained from Sigma-Aldrich (St. Louis, MO, USA).

The sera from four bird species: snowy owl (*Bubo scandiacus*, *n* = 2), Egyptian vulture (*Neophron percnopterus*, *n* = 2), Eurasian griffon vulture (*Gyps fulvus*, n = 2), and Carrion crow (*Corvus corone*, *n* = 4), are part of the Zoo de la Palmyre (Les Mathes, Charente-Maritime, France) serobank used as part of a study designed to estimate the circulation of WNV and USUV flaviviruses in captive avifauna. This study was approved by the Ethics Committee for Clinical Research (ComERC) of Veterinary School of Alfort (EnVA) (Agreement number: 2023-06-23).

### IgG purification
Total IgG from human, bat (*M. myotis*, *M. blythii*, *M. capaccinii* and *N. noctula*), and other mammalian sera was purified using Melon™ Gel IgG Spin Purification Kit (#1859850, Thermo Fischer Scientific, Waltham, MA) according to the manufacturer's protocol, with the sole exception that centrifugation was performed for 30 s instead of 10 s. Concentration of purified IgG was measured by a Nanodrop spectrophotometer (Thermo Fischer Scientific) and its purity was confirmed by SDS-PAGE performed in reducing and non-reducing conditions.

Alternatively, IgG from *M. myotis* and *N. noctula* was purified using protein G agarose (#20397, Thermo Fisher Scientific).

### Purification of bat albumin
Albumin from *M. myotis* was purified from the eluate of Melon™ Gel matrix obtained after exposure of the matrix to PBS after purification of IgG. Samples were diluted in a large volume (5.5 × excess) of binding buffer (50 mM KH$_2$PO$_4$, pH 7.0) and applied on a 1 mL HiTrap Blue HP column (#17041201, Cytiva, Life Sciences, Wilmington, DE) previously equilibrated with binding buffer. The column was washed with 30 mL binding buffer and elution was obtained with a linear gradient of 1.5 M potassium chloride. Albumin purity was confirmed by SDS-PAGE. The bat albumin was biotinylated using EZ-Link® NHS-SS-Biotin (#21441, Thermo Fisher Scientific) following a protocol recommended by the manufacturer. In brief, albumin (19.6 µM) was exposed to 10 × excess of EZ-Link® NHS-SS-Biotin (10 mM stock in DMSO) for 1 h at room temperature. The reaction was stopped by adding 1 M Tris pH 8 to a final concentration of 20 mM.

### Generation of bat F(ab')$_2$ fragments
Purified IgG from *M. myotis* was digested by recombinant IdeS protease from *Streptococcus pyogenes* (kindly provided by Dr. Sébastien Lacroix-Desmazes, Centre de Recherche des Cordeliers, Paris France). The reaction was performed in PBS at a 1:25 (enzyme:substrate) ratio for 16 h at 25 °C. After, the reaction mixture was additionally incubated with protein G agarose (Thermo Fisher Scientific) to remove undigested IgG and Fc fragments. For every 100 µl of the sample, 25 µl of protein G-agarose gel slurry was applied.

### ELISA assays
**Temperature dependence of antigen binding of bat and human IgG.** Thermo Scientific™ Nunc 384-Well Clear Polystyrene Plates (#242765, Thermo Fischer Scientific) were coated overnight at 4 °C (or alternatively for 2 h at room temperature, 22–24 °C) with the following molecules: human Factor VIII (Helixate® FS, CSL Behring, King of Prussia, PA); human Factor IX; human von Willebrand factor (both from LFB, Les Ulis, France); human insulin (#91077C), human hemopexin (#H9291), human hemoglobin (#H7379), porcine thyroglobulin (#T1126), bovine histone III (#H5505), equine cytochrome C (#C2867), lipopolysaccharide from *Escherichia coli* O55:B5, # L2880 (all from Sigma-Aldrich); LysM protein (from *Enterococcus faecalis*, kindly provided by Dr. Stephane Mesnage, University of Sheffield, UK); Asp f1 protein (from *Aspergillus fumigatus*, kindly provided by Dr. Vishu Kumar Aimanianda, Institute Pasteur, Paris); spike protein of SARS-CoV-2 (S protein, Wuhan variant), receptor-binding domain (RBD,

SARS-CoV-2 Wuhan variant), human ACE2 (the three proteins were kindly provided by Dr. Hugo Mouquet, Institute Pasteur, Paris, France); gp120 from HIV-1 (strain CN54, kindly provided by NIH AIDS Reagents Program); rabies virus glycoprotein (#IT-009-001Ep), Ebola Zaire virus glycoprotein (#IT-014-003p), Nipah virus glycoprotein (#IT-026-002p), Hendra virus glycoprotein (#IT-026-001p), NS1 protein of tick-born encephalitis virus (TBEV, #IT-006-0056Ep), measles virus mosaic hemagglutinin (#IT-007-052Ep), HA($\Delta$TM)(A/California/04/2009)

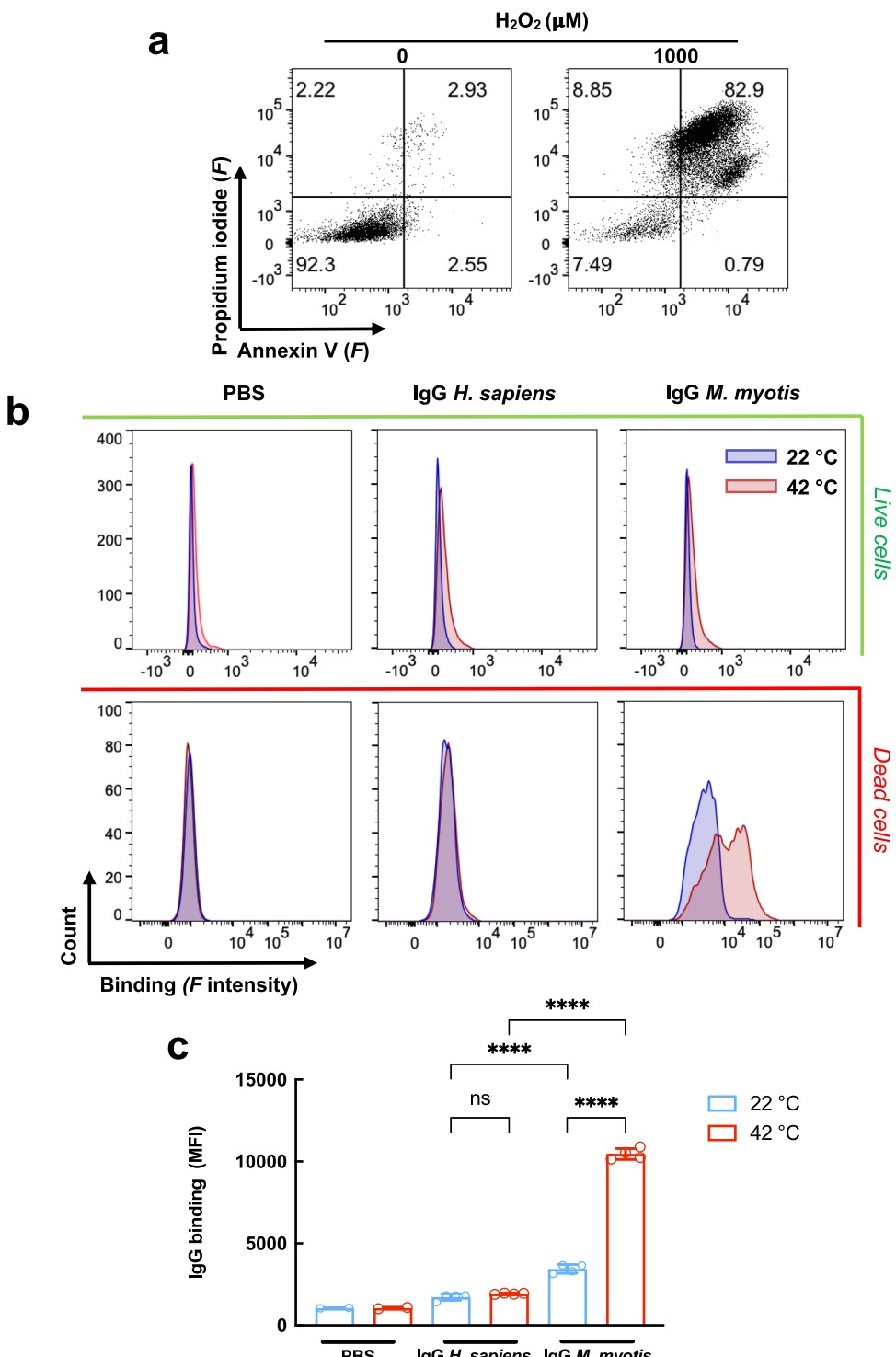

**Fig. 6 | Temperature triggers preferential binding of bat antibodies to dead endothelial cells. a** Gating strategy and identification of cell death induction in HMEC-1 cells by oxidative stress. Dead cells are defined as double positive for annexin V and propidium iodide. **b** Binding of bat (*M. myotis*) and human pooled IgG to alive and dead HMEC-1 cells as a function of temperature. Representative histograms from two independent experiments show binding of fluorochrome labeled (Alexa Fluor™ 555) pooled IgG at 25 µg/ml at 22 °C and 42 °C.

**c** Quantification of bat and human IgG binding to alive and dead HMEC-1 cells. The bars represent average value of MFI ± SD depicting the reactivity of antibodies. The average values were calculated from sample replicates of controls (cells incubated with PBS only, $n = 2$) or the samples (cells incubated with IgG, $n = 4$) of mean fluorescence intensity (MFI). Each circle in the graph represents an MFI value. Statistical significance was evaluated by applying Two-way ANOVA Tukey's multiple comparison test. ****$p < 0.0001$. Source data are provided as a Source Data file.

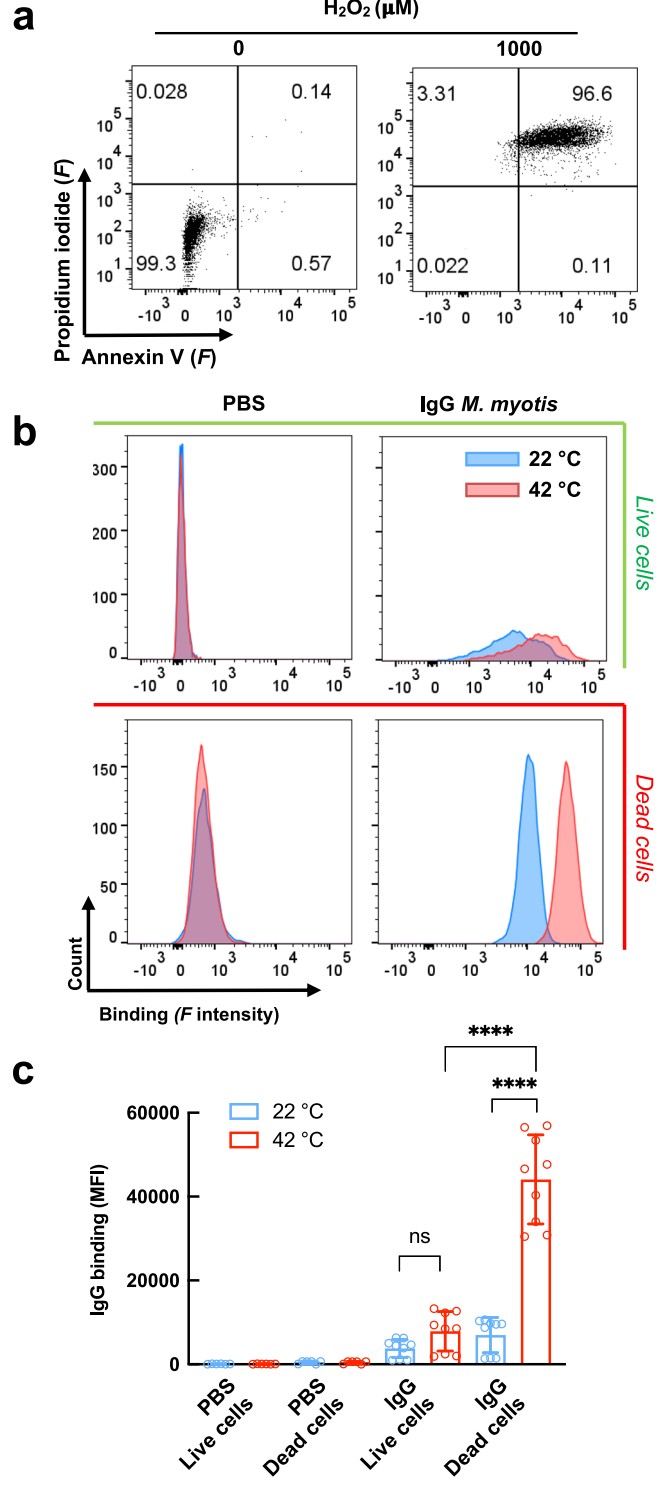

**Fig. 7 | Temperature triggers preferential binding of bat antibodies to dead bat epithelial cells. a** Gating strategy and identification of cell death induction in *M. myotis* epithelial cells (MmNep) by oxidative stress. Dead cells are defined as double positive for annexin V and propidium iodide. **b** Binding of bat (*M. myotis*) IgG to alive and dead MmNep cells as a function of temperature. Representative histograms from four independent experiments shown binding of fluorochrome labeled (Alexa Fluor™ 647) pooled IgG from *M. myotis* at 25 µg/ml at 22 °C and 42 °C. **c** Quantification of bat IgG binding to alive and dead bat epithelial cells. The bars represent average value of MFI ± SD depicting the reactivity of antibodies. The average values were calculated from sample replicates (*n* = 6 for PBS control and *n* = 9 for samples incubated with antibodies) of mean fluorescence intensity (MFI) obtained in three independent experiments. Each circle in the graph represents an MFI value. Statistical significance was evaluated by applying Two-way ANOVA Tukey's multiple comparison test. ****$p$ < 0.0001. Source data are provided as a Source Data file.

applied: 1% solution of bovine serum albumin (#A7030, Sigma-Aldrich) in PBS or 1 × solution of synthetic block (#PA017, Thermo Fisher Scientific) in PBS. Plates were then incubated with: IgG purified from pooled sera of two bat species, *M. myotis* and *N. noctula*; IgG purified from pooled sera of healthy humans; IgG purified from pooled sera of other mammals: mouse (*M. musculus*), rabbit (*O. cuniculus*), goat (*C. hircus*), and cattle (*B. taurus*); and human monoclonal IgG1 antibodies ED38 (highly polyreactive), VRC01 (specific for HIV-1 gp120), and S309 (specific for RBD of SARS-CoV-2), serving as controls. Polyclonal IgG from human, bats, and other mammals was typically diluted to 0.07 mg/ml (or to 0.02 or 0.1 mg/ml) in PBS containing 0.05% Tween 20 (T-PBS). Monoclonal IgG antibodies were diluted to concentrations of 0.01 and 0.07 mg/ml in T-PBS. In a specific experimental setting, reactivity of IgG in whole sera from *M. myotis* or human were tested. To this end, sera were diluted 200 × in T-PBS. In another experiment, plates were incubated with sera from different species of birds: snowy owl (*B. scandiacus*), Egyptian vulture (*N. percnopterus*), Eurasian griffon vulture (*G. fulvus*), and Carrion crow (*C. corone*), diluted 100 × in T-PBS. The purified polyclonal IgG from bats, humans, other mammals or monoclonal human IgG1 antibodies as well as diluted sera from human, bats and birds, were incubated for 2 h with immobilized antigens in parallel at three different temperatures: 4 °C, 22 °C, and 42 °C (one plate per temperature). After, the microtiter plates were washed 5 × with T-PBS and then incubated for 1 h at room temperature with 2 µg/ml of biotinylated recombinant protein G (#29988, Pierce™, Thermo Fisher Scientific) for recognition of IgG antibodies from human, bats, and other mammalian species. Following thorough washing, streptavidin conjugated with HRP (Invitrogen, Thermo Fisher Scientific) diluted 100× was added and incubated for 30 min at room temperature. For detection of bird's IgY antibodies, plates were incubated with goat anti-chicken IgY-HRP (#6100-05, Southern Biotech, AL, USA), diluted 3000 × in T-PBS. This antibody showed high cross-reactivity for IgY from different bird species. After thorough washing with T-PBS, the reactivity of antibodies was revealed by adding peroxidase substrate, *o*-phenylenediamine dihydrochloride (#P8287, Sigma-Adrich). After stopping the reaction with 2 M HCl, the absorbance was measured with a microplate reader (Infinite 200 Pro, Tecan, Männedorf, Switzerland) at 492 nm.

(H1N1, # IT-007-052Ep), hepatitis D virus small Ag (HDVAg, #IT-004-021Ep), Dengue virus 2 Capsid Protein (#IT-006-015Ep), Chikungunya virus Capsid protein (#IT-022-003Ep), HPV-16 L2 protein (#IT-027-012Ep), Antigen Mpt70 (*M. tuberculosis*/H37Rv, #IT-500-012Ep), Antigen 85-B (*M. tuberculosis*/H37Rv, #IT-500-015Ep), and p47 (*T. pallidum*/Nichols strain, #IT-501-004Ep). All viral and bacterial proteins listed afterwards gp120 were obtained from Immune Technology Corp. (New York, NY, USA). Plates were incubated with proteins usually diluted to 5 µg/ml in PBS; insulin and LPS were diluted at 10 µg/ml. After the coating step, plates were blocked by incubation for 2 h with PBS containing 0.25% Tween 20. Alternatively, other blocking agents were

**Titration of bat and human sera.** NS1 protein of tick-born encephalitis virus (TBEV), rabies virus glycoprotein, influenza hemagglutinin, and SARS-CoV-2 S protein were immobilized at 2 µg/ml on 96-well polystyrene microtiter plates Nunc MaxiSorp™ (#442404, Thermo Fisher Scientific). Following blocking with 1× synthetic block (Thermo Fisher Scientific) in PBS, whole sera from human and *M. myotis* was serially diluted (starting from 60 × to 1920×) in T-PBS and incubated for 2 h at 4 °C, 22 °C, 42 °C with immobilized antigens (human serum with SARS-CoV-2 S protein and influenza A hemagglutinin; bat serum with TBEV NS1 and rabies virus glycoprotein). Detection of bound bat IgGs was

done by incubation for 1 h at room temperature with mouse anti-bat immunoglobulin (#NBP2-23483, L-chain specific, clone BT1-4F10, Novus Biologicals, CO, USA) diluted to 1 µg/ml in T-PBS followed by incubation for 1 h with goat-anti mouse IgG conjugated with HRP (Southern Biotech) diluted 2000× in T-PBS. Binding of human IgG was detected by 1 h incubation at room temperature with anti-human IgG, conjugated with HRP (#9040-05, clone JDC-10, Southern Biotech) diluted 3000× in T-PBS. The succeeding steps of the experiment were identical to those described for the previous ELISA setting.

**Comparison of the reactivity of bat albumin and bat IgG with crude extract of bat epithelial cells.** Total lysate of the nasal epithelial cell line (MmNep)[67] of *M. myotis* was prepared by freezing thoroughly washed with PBS cells ($6.3 \times 10^7$ cells in 1 ml PBS). After thawing, the suspension was immediately sonicated and centrifuged at $11,000 \times g$ for 10 min. The total protein concentration of the supernatant was determined by Nanodrop. The crude extract was coated on Nunc 384-Well Clear Polystyrene Plates (Thermo Fischer Scientific) at a concentration of 20 µg/ml in PBS for 2 h at room temperature. Plates were blocked by a solution of 0.25% of Tween 20 in PBS. Purified IgG and biotinylated albumin from *M. myotis*, were incubated at serial dilutions ranging 200–3.125 nM in T-PBS for 1 h. Identical samples were incubated at 4 °C, 22 °C, and 42 °C. After washing, plates were incubated for 1 h with 2 µg/ml biotinylated protein G in T-PBS for detection of bat IgG. The IgG and albumin reactivity to cell antigens was detected by incubation for 1 h with streptavidin-HRP (Thermo Fischer Scientific) diluted 100× in T-PBS. The succeeding experimental steps were identical to the principal ELISA experiment described above.

**Temperature sensitivity of antigen binding of F(ab')$_2$ fragments from bat IgG.** Thermo Scientific™ Nunc 384-Well Clear Polystyrene Plates (Thermo Fischer Scientific) were coated for 2 h at room temperature with 5 µg/ml of rabies virus glycoprotein, NS1 protein of tick-born encephalitis virus (TBEV), measles virus mosaic hemagglutinin, Antigen Mpt70 (*M.Tuberculosis*/H37Rv), and p47 (*T. pallidum*/Nichols strain) in PBS. Plates were blocked with PBS containing 0.25% Tween 20 for 1 h at room temperature followed by incubation for 1 h with 10 µg/ml of F(ab')$_2$ fragments from *M. myotis* in parallel at 4 °C, 22 °C and 42 °C. Detection of bound bat F(ab')$_2$ fragments was done by incubation for 1 h at room temperature with mouse anti-bat immunoglobulin (L-chain specific, clone BT1-4F10, Novus Biologicals, CO, USA) diluted 1 µg/ml in T-PBS followed by incubation for 1 h with goat-anti mouse IgG conjugated with HRP (Southern Biotech) diluted 2000× in T-PBS.

**Ionic strength dependence of binding of human and bat IgG.** Half-area 96-well polystyrene plates (#675061, Greiner-Bio-One, Les Ulis, France) were coated by incubation for 2 h at room temperature with 2 µg/ml of NS1 protein of tick-born encephalitis virus (TBEV), rabies virus glycoprotein, influenza hemagglutinin, measles virus mosaic hemagglutinin, SARS-CoV-2 S protein, and LysM from *E. faecalis* in PBS. After, plates were blocked by 1 h incubation with a solution of PBS containing 1× synthetic block (Thermo Fisher Scientific) and 0.25% Tween 20. Purified human and bat (*M. myotis* and *N. noctula*) IgG was diluted to 20 µg/ml in 10 mM HEPES pH 7.5, 0.05% Tween 20 buffer containing increasing concentrations NaCl: 0, 0.0078–4 M, then it was incubated for 2 h at 42 °C. Bat antibodies were incubated with: TBEV NS1, measles virus hemagglutinin, and rabies virus glycoproteins; human antibodies were incubated with: influenza hemagglutinin, SARS-CoV-2 S protein, and LysM from *E. faecalis*. Following steps were as in the principal ELISA experiment described above.

**Interaction of bat and human IgG with protoporphyrin IX.** Protoporphyrin IX (#P8293, Sigma-Alrich) was covalently conjugated on the surface of gelatin pre-coated half-area 96-well polystyrene plates

(Greiner-Bio-One) following an amino-coupling procedure previously described for heme[68]. The hapten-coated plates were blocked for 1 h with 0.25% solution of Tween 20 in PBS. Purified pooled IgG from bat (*M. myotis*) and human were serially diluted in the range 100–1.562 µg/ml (670–10.468 nM) in T-PBS and incubated for 90 min at room temperature. Next steps for the detection of bound IgG were as in the principal ELISA experiment described above.

**Assessment of polyreactivity of IgG antibodies.** To assess the level of polyreactive antibodies in polyclonal bat and human IgG samples, we used an established assay based on the analyses of antibody reactivity to a nitroarene hapten[34]. Ninety-six well polystyrene plates NUNC MaxiSorp™ or NUNC 384-Well Clear polystyrene plates (Thermo Fisher Scientific) were coated with 2,4-dinitrophenyl conjugated to BSA (#A6661, DNP-BSA, Sigma-Aldrich), diluted to 10 µg/ml in PBS and incubated for 2 h at room temperature (22–24 °C). As a control, unconjugated BSA was coated under identical conditions. Plates were blocked by saturation with a 0.25% solution of Tween 20 in PBS for 1 h at room temperature. For evaluation of antibody reactivity to DNP, IgG purified from human and bat (*M. myotis* and *N. noctula*) pooled sera was diluted to 670 nM (100 µg/ml) in PBS-T, followed by two-fold serial dilution to 10.468 nM (1.56 µg/ml). Monoreactive (VRC01 and S309) and polyreactive (ED38) monoclonal IgG1 antibodies were used as controls. These were first diluted to 400 nM (60 µg/ml) followed by threefold serial dilutions to 0.0068 nM (0.001 µg/ml) in PBS-T. Polyclonal and monoclonal antibodies were incubated on microtiter plates for 1 h at room temperature. Following steps were as described above.

**Real-time interaction analyses−temperature dependence of antigen binding of bat IgG**
The interaction of bat IgG with selected viral proteins was evaluated by a surface plasmon resonance-based system−Biacore 2000 (Cytiva, Biacore, Uppsala, Sweden). Recombinant rabies virus glycoprotein and NS1 protein of TBEV were covalently immobilized on CM5 sensor chips (#29149604, Cytiva, Biacore) using an amine-coupling kit (#BR100050, Cytiva, Biacore) according to the procedure recommended by the manufacturer. In brief, viral proteins were diluted to a final concentration of 10 µg/ml in 5 mM maleic acid, pH 4, and injected over pre-activated sensor surfaces. Carboxyl groups of the carboxymethylated dextran were activated by exposure of the surface for 4 min to a mixture of 1-ethyl-3-(3-dimethylaminopropyl)-carbodiimide and N-hydroxysuccinimide. For inactivation of the activated carboxyl groups that were not engaged in interactions with the amino groups from the proteins, sensor surfaces were exposed for 4 min to 1 M solution of ethanolamine.HCl. Covalent immobilization of viral proteins and all kinetic measurements were performed with HBS-EP (10 mM HEPES pH 7.2; 150 mM NaCl; 3 mM EDTA, and 0.005% Tween 20) used as running and sample dilution buffer. Prior to real-time binding analyses, the running buffer was always filtered through a 0.22 µm filter and degassed under a vacuum.

To measure interactions of bat antibodies with viral proteins as a function of temperature, IgG purified from a serum pool from *M. myotis* was serially diluted in the running buffer in the range 100–6.25 µg/ml (670–41.875 nM) and injected over the sensor surface. The flow rate during all interaction analyses was set at 30 µl/min. The association and dissociation phases of bat polyclonal IgG binding were monitored for 4 and 5 min, respectively. Between consecutive sample injections, the sensor chip surface was regenerated by a brief exposure (30 s) to a solution of 2 M NaSCN (#467871, Sigma-Aldrich). All analyses were performed at the following temperatures: 4 °C, 6 °C, 16 °C, 24 °C, 32 °C, and 40 °C. Real-time binding curves were obtained after subtracting bat IgG binding to a reference surface that lacked immobilized viral proteins. Binding curves obtained at 40 °C allowed reliable evaluation of binding kinetics. To this end, we used BIAevaluation software v. 4.1.1 (Cytiva, Biacore) performing global kinetics analyses

applying the Langmuir binding model. The apparent binding kinetics and affinity were evaluated under the assumption that 10% of all antibodies in the pooled preparation were able to bind viral proteins.

### Real-time interaction analyses−temperature dependence of Fc-dependent interactions of bat and human IgG

Purified bat (*M. myotis*, n = 10; and *N. noctula*, n = 35), and human pooled IgG were immobilized on CM5 sensor chips (Cytiva, Life Sciences, Biacore). To this end, IgG samples were diluted to a final concentration of 10 μg/ml in 5 mM maleic acid, pH 4, and injected over pre-activated sensor surfaces. Other parameters of the immobilization procedure were identical as described above.

To assess the temperature-dependence of Fc-mediated interactions of IgG, the binding of human C1q (#A099, Comptech, TX, USA), recombinant human FcRn (kindly provided by Dr. Sune Justesen, Immunitrack Aps, Copenhagen, Denmark), and recombinant protein G (Pierce™, Thermo Fisher Scientific) was elucidated. The real-time interaction analyses with C1q and protein G were performed in HBS-EP buffer. The analyses of the binding of FcRn to IgG were studied in a buffer containing 100 mM NaCl, 100 mM Tris pH 5.7 (adjusted with dry citric acid), and 0.1% Tween-20. C1q was injected after dilution to 0.625 nM; FcRn was injected at serial dilutions in the range: 25−0.097 nM (2 × dilution step); the binding of protein G was measured at serial dilutions in the range: 100−0.046 nM (3 × dilution step). Buffer flow rate during all interaction analyses was set at 30 μl/min. The association and dissociation phases of all interactions were monitored for 4 and 5 min, respectively. Between the injections of Fc-binding proteins, the sensor chip surface was regenerated by a brief exposure (30 sec) to a solution of 1.5 M NaSCN (in the case of C1q); 100 mM NaCl, 100 mM Tris with pH 7.8 (in the case of FcRn), and a solution of 2 M NaOH and 0.1 M NaOH (in the case of protein G). The real-time binding measurements were performed as a function of temperature at 4 °C, 22 °C and 40 °C (for C1q) and at 22 and 40 °C (for FcRn and protein G). For data analyses BIAevaluation software v. 4.1.1 (Cytiva, Biacore) was used for performing global kinetics analyses applying the Langmuir binding model.

### Immunoblot analyses

Cell lysate of MmNep cells (epithelial cells, *Myotis myotis*)[67] or *Bacillus anthracis* were loaded to 1-well 4−12% Bis-Tris NuPAGE Novex SDS-PAGE gels (#NP0324BOX, Invitrogen, Thermo Fisher Scientific). Proteins were separated for 45 min at 200 V. After migration, they were transferred to nitrocellulose membranes using an iBlot electrotransfer system (Invitrogen, Thermo Fisher Scientific). Membranes were blocked overnight with PBS containing 0.1% Tween 20 (T-PBS). Next, they were conditioned in PBS 1/3 × + 0.1% Tween 20 for 10 min and fixed on a Miniblot system (Immunetics, Cambridge, MA). They were then incubated for 1 h at 22 °C with IgG from *M. myotis (*n = 10 individuals*)* and *N. noctula* (*n* = 35 individuals) in twofold dilutions, from 20 μg/ml to 2.5 μg/ml, in phosphate buffer (PBS) with NaCl concentration of eighter 150 mM or 50 mM.

Membranes were washed (6 × 10 min) with T-PBS and then incubated for 1 h at room temperature with 2 μg/mL biotinylated Protein-G (Thermo Fisher Scientific) in T-PBS. Next, they were washed (6 × 5 min) with T-PBS 0.1% and incubated with streptavidine coupled with alkaline phosphatase (Southern Biotech) diluted 2000×. Finally, membranes were washed (6 × 5 min) with T-PBS 0.1%, rinsed with water to remove phosphate ions, and revealed by incubation with SigmaFast BCIP/NBT substrate solution (#B5655, Sigma-Aldrich). They were then left to dry for 24 h and images were taken on the iBright CL1500 Imaging System (Thermo Fisher Scientific).

### Conjugation of human and bat IgG with fluorochromes

Human and bat antibodies were covalently labeled with fluorochromes via amine-coupling reaction. First, polyclonal IgG purified from pooled human and bat (*M. myotis*) sera were diluted to 2 μM in a solution consisting of 1/3 100 mM carbonate buffer, pH 9, and 2/3 PBS. Amine-reactive Alexa Fluor™ 555 NHS ester (#A37571, Thermo Fisher Scientific) was added to IgG samples to a final concentration of fluorochrome 40 μM and incubated for 1 h at room temperature with constant gentle agitation. Additionally, human and bat IgG was labeled with Alexa Fluor™ 647 NHS ester (#A37573, Thermo Fisher Scientific) under identical conditions. Following the conjugation reaction, we separated labeled IgG from unconjugated fluorochromes by applying PD-10 columns with Sephadex G-25 resin (#17085101, Cytiva, Marlborough, MA). Conjugation efficacy and conjugated IgG concentration were measured by absorbance spectroscopy using an Agilent Cary 100 UV-vis spectrophotometer (Agilent Technologies, Santa Clara, CA).

### Immunofluorescence analyses

To evaluate the reactivity of bat (*M. myotis, M. blythii, M. cappaccinii, and N. noctula*) and human IgG to HEp2 cells, we used the AESKUS-LIDES® ANA-HEp-2 diagnostic kit for detection of autoantibodies recognizing cellular antigens by immunofluorescence (#51.100, Aesku Diagnostics, Wendelsheim, Germany). We followed the protocol recommended by the manufacturer with certain modifications. Thus, human and bat IgG conjugated to Alexa Fluor™ 555 was diluted to 50 μg/ml in PBS and incubated for 30 min in a moist chamber with Hep-2 cells pre-fixed on glass slides. To evaluate the reactivity of bat (*M. myotis*) IgG to bat nasal epithelial cells, we cultivated MmNep[67] cells in complete DMEM/F-12+GlutaMAX™ (#10565018, Gibco™, Thermo Fisher Scientific) containing 10% fetal calf serum (#30-2021, ATCC) and 1% Penicillin-Streptomycin (#A5873601, Gibco™, Thermo Fisher Scientific) on coverslips (10 mm diameter). After reaching 80% confluence, the cells were fixed with 4% PFA mixed with Hoechst dye (#H3570, Thermo Fisher Scientific) diluted 5000×. The coverslips were thoroughly washed after each following step. The cells were permeabilized with a 0.2% Triton-X-100 solution (#T8787, Sigma-Aldrich). Next, they were incubated for 30 min with bat IgG coupled with Alexa Fluor™ at 25 μg/ml. Finally, the cells were incubated with Phalloidin-iFluor 488 diluted 1000 × (#Phalloidin-iFluor 488, Abcam, Cambridge, United Kingdom) for 20 min. The Hoechst and Phalloidin staining were used to facilitate cell counting.

To evaluate the effect of temperature on IgG binding, slides with identical samples were incubated at 22 °C and 42 °C. As a negative control, some cells were incubated with PBS only or with 50 μg/ml of Alexa Fluor™ 555-streptavidin (Invitrogen, Thermo Fisher Scientific). After washing with PBS and applying mounting medium and cover slips, immunofluorescence was read by Zeiss Axio Observer Z1 microscope (Carl Zeiss AG, Oberkochen, Germany) equipped with ApoTome system. The acquisition time was 80 ms (HEp-2 cells) or 50 ms (MmNep cells). Acquired fluorescence images with magnifications of 63× were colored digitally in green. The acquisition and data analyses were performed with ZEN software (Carl Zeiss AG). Quantification of mean fluorescence intensity (MFI), reflecting antibody interaction with the cells, was normalized by the number of cells observed in the bright field.

### Protein microarrays

To evaluate the effect of temperature on the immunoreactivity of human and bat (*M. myotis and N. noctula*) polyclonal IgG to a broad repertoire of human proteins (>17,000), we applied HuProt™ v4.0 protein microarrays (CDI Laboratories, Inc., Baltimore, MD). First, microarray slides were blocked by incubation for 2 h at room temperature with PBS containing 1 × synthetic block (Thermo Fisher Scientific) and 0.05% Tween 20. After washing for 5 min with PBS containing 0.05% Tween 20 (PBS-T), chips were incubated for 2 h at 22 °C or 42 °C with Alexa Fluor™ 647-conjugated human and *M. myotis* IgG (for 22 °C) and Alexa Fluor™ 555-conjugated human and *M. myotis* IgG (for 42 °C) with constant agitation on an orbital shaker (at 50 rpm).

Alexa Fluor™ 555-conjugated IgG from *N. noctula* was used for analyses at both 22 and 42 °C. IgG concentration was 5 μg/ml in 5 mL PBS-T. Following incubation with antibodies, microarray slides were washed 3 × 5 min with PBS-T with constant agitation (50 rpm). After, salts were removed by soaking the slides in 1/10 solution of PBS in deionized water and dried by centrifugation (100 × *g*) for 1 min. Fluorescence intensity at 532 nm and 635 nm was measured by the microarray scanner GenePix 4000B (Molecular Devices, San Jose, CA). Raw data from microarray chips were quantified and analyzed using Spotxel software v. 1.7.7 (Sicasys, Heidelberg, Germany).

## Thermal shift assay for estimation of thermodynamic stability of IgG
To assess the thermodynamic stability of human and bat (*M. myotis* and *M. capaccinii*) purified IgG molecules, we used Protein Thermal Shift Dye Kit (#4461146, Applied Biosystems, Thermo Fisher Scientific) following the manufacturer's protocol. All IgG samples were diluted in PBS and tested at a concentration of 80 μg/ml. Protein unfolding reactions and fluorescent signal detection were performed in Applied Biosystems 7900 HT Fast Real-Time PCR machine (Applied Biosystems, Thermo Fisher Scientific). For data analyses, we used GraphPad Prism v. 9.5 software (GraphPad Software, San Diego, CA). Melting points ($T_m$) of IgG molecules were calculated by fitting experimental data using the Boltzmann equation.

## Hydrophobic interaction chromatography
Overall hydrophobicity of human and bat (*M. myotis*) IgG was assessed by hydrophobic interaction chromatography. To this end, 70 μg of purified IgG were diluted in 1.1 M solution of ammonium sulfate and injected into a 1 mL HiTrap Octyl FF column (#17135901, Cytiva, Life Sciences). The column was previously equilibrated with binding buffer (50 mM sodium phosphate, 1.1 M ammonium sulfate, pH 7.0). It was further washed with 5 mL binding buffer and elution was performed with a linear gradient of low ionic strength buffer (50 mM sodium phosphate, pH 7.0) for 20 × column volumes. Chromatograms for both species were exported and overlayed with GraphPad Prism v. 9.5 software (GraphPad Software, San Diego, CA).

## Flow cytometry analyses
**Human endothelial cells.** Human skin endothelial cell line HMEC-1 (#CRL-3243, ATTC, Manassas, VA) was cultured under standard conditions in a complete MCDB 131 cell culture medium (#10372019, Gibco™, Thermo Fisher Scientific) containing 10% fetal calf serum (ATCC), 10 ng/mL epidermal growth factor (#PHG0314, EGF, Thermo Fisher Scientific), 1 pg/mL Hydrocortisone (Sigma-Aldrich), and 10 mM glutamine (ATCC). To create pro-apoptotic conditions, HMEC-1 cells were first starved in a serum-free medium Opti-MEM™ (#31985062 Gibco™, Thermo Fisher Scientific) for 24 h, then they were exposed for 24 h to Opti-MEM™ containing 1 mM of $H_2O_2$. To represent normal conditions, MCDB 131 medium was not changed. After incubation, the cells were washed with PBS by centrifugation at 125 × *g* for 10 min at 4 °C. A sample of cells was stained with Trypan Blue (#15250061, Gibco™, Thermo Fisher Scientific) to estimate the percentage of dead cells. Following washing with PBS, HMEC-1 cells were resuspended in PBS and distributed in FACS tubes with $10^5$ cells per tube. For assessment of cell death by flow cytometry, a part of the samples was labeled with annexin V-conjugated with APC and propidium iodide (#P1304MP, Thermo Fisher Scientific). HMEC-1 cells cultured in pro-apoptotic and normal conditions were distributed and incubated in the presence of Alexa Fluor™ 555-conjugated polyclonal human and bat (*M. myotis*) IgG at a concentration of 25 μg/ml. To assess the effect of temperature on IgG binding to live and dead cells, samples were incubated for 30 min at 22 °C or at 42 °C. Cells were then washed with PBS at 125 × *g*, 4 °C for 10 min and binding was analyzed by flow

cytometer FACS LSRFortessa™ (BD Bioscience, Franklin Lakes, NJ) using Yellow/Green Ex 561 nm Em 586/15 laser for the detection of antibodies. Mean fluorescence intensity (MFI) indicating antibody binding was measured using FlowJo Software (LLC, Ashland, OR). The data was normalized by calculating the MFI ratio against the mean negative control.

**Bat epithelial cells.** The *Myotis myotis* nasal epithelial cell line MmNep[67] was cultured in complete DMEM/F-12+GlutaMAX™ (Gibco™, Thermo Fisher Scientific) containing 10% fetal calf serum (ATCC) and 1% Penicillin–Streptomycin. To create pro-apoptotic conditions, MmNep cells were first starved in a serum free DMEM/F-12+GlutaMAX™ (Gibco™, Thermo Fisher Scientific) for 24 h, then they were exposed for 24 h to DMEM/F-12+GlutaMAX™ containing 1 mM of $H_2O_2$. To represent normal conditions, DMEM/F-12+GlutaMAX™ medium was not changed. After incubation, the cells were detached by Trypsin 0.25%, washed with PBS by centrifugation at 125 × *g* for 10 min at RT and resuspended in PBS. A sample of cells was stained with Trypan Blue (Gibco™, Thermo Fisher Scientific) to estimate cell number and condition. Following washing with PBS, MmNep cells were placed in FACS tubes with $3 × 10^5$ cells per tube.

MmNep cells cultured in pro-apoptotic and normal conditions were distributed and incubated in the presence of Alexa Fluor™ 647-conjugated pooled bat (*M. myotis*) IgG at a concentration of 25 μg/ml. To assess the effect of temperature on IgG binding to live and dead cells, samples and controls were incubated for 30 min at 22 or 42 °C. Cells were then washed with PBS at 125 × *g*, 4 °C for 10 min. For assessment of cell death by flow cytometry, control samples were labeled with annexin V-conjugated with FITC and propidium iodide (Molecular Probes, Thermo Fisher Scientific). Binding was analyzed by flow cytometer FACS LSR 2 (BD Bioscience, Franklin Lakes, NJ) using Red 660/20 for the detection of antibodies. Mean fluorescence intensity (MFI) indicating antibody binding was measured using FlowJo Software (LLC, Ashland, OR).

## Bioinformatics analyses
Sequences of *M. lucifugus* variable regions (VH3) were retrieved from the literature[30]. Sequences of human variable regions (VH3) from naïve and IgM memory B cells were obtained from following study[69]. Framework (FR) and Complementarity-Determining (CDR) regions were delineated based on IMGT unique numbering[70]. Of note, the three first amino acids at the N-terminal extremity of the FRH1 were not provided for the bat sequences. Accordingly, these residues were also removed from the analysis of human sequences. GRAVY index was computed with Gravy Calculator (https://www.gravy-calculator.de). Amino acid counts were done with Microsoft Excel (version 2018). Statistical analysis was performed with GraphPad Prism v. 9.5 software.

Sequences of bat constant region were obtained from NCBI Genome Data Viewer[71] selecting *Myotis myotis* organism and searching for loci annotated as "immunoglobulin heavy constant gamma". Among the three hits, we selected those localized on NW_023416326.1 and NW_023416348.1 (the sequence of the third hit is identical to the sequence on NW_023416326.1). Human and mouse reference sequences for the constant region of heavy chains were obtained from IMGT/GENE-DB[72]. Sequence alignments, phylogenetic trees, and percent of identity were obtained with Clustal Omega[73].

## Reporting summary
Further information on research design is available in the Nature Portfolio Reporting Summary linked to this article.

# Data availability
Source data are provided with this paper. All additional raw data supporting the findings of this study are available upon request from the corresponding author. Source data are provided with this paper.

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

## Acknowledgements

The experimental work was supported by a grant from European Research Council (Project CoBABATI ERC-StG-678905), awarded to J.D.D. Fieldwork funding was provided by the Bulgarian National Science Fund (project KP-06-N51/9 "Caves as a reservoir for novel and reoccurring zoonoses—ecological monitoring and metagenomic analysis in real-time"). N.T. was funded by a Ph.D. Fellowship from Karoll Knowledge Foundation and a stipend from the French Institute, Bulgaria. V.Z. was supported by a stipend from the French Institute, Bulgaria. A.R.R. is recipient of a fellowship from an Innovative Training Network (ITN) funded by the European Union's Horizon 2020 Programme under grant agreement No 859974, project EDUC8.

We would like to express our gratitude to the volunteers who assisted us in the field: Boris Petrov, Petar Velkov, Katrin Dimitrova, Kaloyana Koseva, Kristin Meshinska, and Stela-Teodora Trendafilova. Our heartfelt thanks go to Dr Lubka Roumenina, and Dr Sebastien Lacroix-Desmazes (both from Centre de Recherche des Cordeliers, Paris, France) for their insightful feedback on our ideas and their unwavering intellectual support throughout this study. We thank Dr Hugo Mouquet, Dr Vishu Kumar Aimanianda (both from Institute Pasteur, Paris, France) and Dr Stephane Mesnage, (University of Sheffield, UK) for providing us with some important material (pathogen proteins, antibodies, and plasmids). We also thank Pr. J. Pikula and Dr. V. Kovácová (Department of Ecology and Diseases of Zoo Animals, Game, Fish and Bees, University of Veterinary and Pharmaceutical Sciences Brno) for sharing the MmNep cells.

## Author contributions

N.T., V.Z., and J.D.D. conceived the study; N.T., V.Z., N.J., C.P., and J.D.D., designed the experiments; N.T. and V.Z. collected essential samples; N.J. and G.G. provided essential samples and materials; N.T., V.Z., A.R.R., E.H., M.C.D., R.V.L., M.L. and J.D.D. performed the experiments. N.T., V.Z., A.R.R., E.H., R.V.L., M.L., and J.D.D. analyzed the obtained data; J.D.D. supervised the research; J.D.D. wrote the manuscript; all authors contributed to editing the manuscript.

## Competing interests

The authors declare no competing interests.
