## [Peer Review File · Nature Communications]

Temperature sensitivity of bat antibodies links metabolic state of bats with antigen-recognition diversityREVIEWER COMMENTS

Reviewer #1 (Remarks to the Author):

This is a very interesting and well written manuscript from the Dimitrov lab. Overall, it is remarkable that bats exploit the temperature-dependent gain in antibody polyreactivity to such a dramatic extent. I believe that the paper will be appealing to a wide audience from immunologists to antibody drug developers. However, some basic insights about the observed temperature-dependent effects are currently missing from the manuscript:

- What are the physicochemical traits contributing to the higher polyreactivity of bat antibodies? Is there sequencing data of bat antibody repertoires that reveals unusually high usage of amino acids and motifs linked to polyreactivity? Should we expect specific sequence motifs in bat antibodies that cause the strong temperature-dependent polyreactivity?
- The temperature-dependent binding is a known feature of polyreactive antibodies. Can the authors comment (supported by references or data) on whether bats generally have more polyreactive antibodies compared to humans? The authors mention something in this direction on lines 329 and 330, but it seems that this is a statement based only on differences in the temperature-dependent polyreactivity.
- Can you confirm that the gain in polyreactivity at 42 °C is due to binding via the variable antibody domains and not because of an unexpected non-specific interaction via the constant antibody domains of bats?
- The CH2 domains of human IgGs unfold at about 60-70 °C. Can you confirm that the measured melting temperatures are due to thermodynamic stability differences of the Fabs and not the constant antibody domains?
- ANS was used to probe the hydrophobicity of proteins, but an orthogonal assay (retention times from hydrophobic interaction chromatography) can support the claim for unusually high hydrophobicity of bat IgGs.

Minor:

- Some abbreviations are not spelled out at their first mention in the manuscript (e.g., IFN, DAMP). These are standard abbreviations in immunology but not clear for a broad readership.

Reviewer #2 (Remarks to the Author):

The authors' work is very intriguing, which is important in itself for understanding bat immunity, and which might also provide insight into engineering efforts of temperature-sensitive antibodies. But I feel there is a lack of sufficient depth and quality to this work. Here are my specific comments:

Major comments

- (1) The authors' observation in Fig. 1a is certainly very interesting. To corroborate the observations and to make sense of their impact in a broader context, can the authors perform the same study with total IgG from some bird species (flying but not mammals) and from mouse (mammals that don't fly)?
- (2) Not sure why the authors focus on antibodies. What about T cell receptors of bats? Do they have the same temperature-dependent trait? More broadly, is it possible that all bat proteins (not antibodies, not TCRs) all have this thermodynamic instability? Then all of them are just broken proteins that are sticky at high temperature? Knowing this will help shape more accurate interpretation of the findings of the authors
- (3) I think the authors need to dive deep into why the bat antibodies would demonstrate temperature-dependent affinity properties, to give the work sufficient depth. Even just bioinformatics analyses of antibody sequence/structure features would be very helpful, as long as such analyses can provide some novel insight
- (4) The observation with endothelial cells is interesting and looks real. But cells of many other types could also die during inflammation induced by flight. What about opsonization of these cell types? Uncertain why the authors focused on endothelial cells

(5) "these data indicate that bat antibodies acquire preferential reactivity to dead 240 endothelial cells at elevated temperatures" What makes the dead cells different from live cells so that bat antibodies would react to them only at high temperature?

Minor comments

(1) "The experiment 188 demonstrated that bat IgG was characterized by a substantially higher hydrophobicity as 189 compared to human IgG." Will this be the cause of the temperature-dependent binding properties of bat antibodies? The authors didn't explain the meaning of this finding.

Reviewer #1 (Remarks to the Author):

This is a very interesting and well written manuscript from the Dimitrov lab. Overall, it is remarkable that bats exploit the temperature-dependent gain in antibody polyreactivity to such a dramatic extent. I believe that the paper will be appealing to a wide audience from immunologists to antibody drug developers. However, some basic insights about the observed temperature-dependent effects are currently missing from the manuscript:

We are thankful to the Reviewer for the positive evaluation of our work.

- What are the physicochemical traits contributing to the higher polyreactivity of bat antibodies? Is there sequencing data of bat antibody repertoires that reveals unusually high usage of amino acids and motifs linked to polyreactivity? Should we expect specific sequence motifs in bat antibodies that cause the strong temperature-dependent polyreactivity?

We concur with the Reviewer's suggestion that comparing the sequences of the variable regions of human and bat antibodies could yield valuable insights into the molecular attributes responsible for the heightened polyreactivity and temperature sensitivity observed in bat antibodies. Achieving this necessitates obtaining full sequences of the variable regions post V(D)J recombination for a substantial number of antibodies. As far as we are aware, however, comprehensive sequence data regarding antibody repertoires with fully assembled V(D)J genes for heavy and light chains are not available for the bat species under investigation in our study or for other bat species. Nevertheless, this query could be addressed by leveraging data from a previous immunogenetics study (PMID: 20547175), where the V_{H3} gene families of a bat species, *Myotis lucifugus*, were annotated. Notably, *M. lucifugus* belongs to the same genus (*Myotis*) as three of the species examined in our research.

To address Reviewer's query, we utilized the published sequence information and juxtaposed the sequences of the V_{H3} gene family with those of human germline (or near germline) V_{H3} genes. Comparative analyses revealed no significant differences in the overall organization and sizes of subregions encoded by the V_H genes (framework regions 1-3 and CDR H1 and H2). However, computation of the overall hydrophobicity using the GRAVY tool unveiled a notably higher level of hydrophobicity in V_H genes from *Myotis* compared to human V_H genes. This increased hydrophobicity primarily stemmed from higher hydrophobicity in framework region FR1 and CDRH2. These findings align with our results, which demonstrate the involvement of hydrophobic effects in antigen binding by antibodies from *Myotis myotis* (refer to Fig. 3c in the revised manuscript). Moreover, they correlate with the heightened polyreactivity and thermo-sensitivity, as previous studies have shown that hydrophobic interactions strengthen with rising temperatures. However, these data should be interpreted with caution, as the analyzed sequences represent only a portion of the variable region, lacking CDRH3 and the joining segment.

We have now incorporated the newly acquired in silico data into the manuscript as a novel panel (Fig. 4d) and have appropriately described and discussed the results within the text.

In addition to comparing variable gene sequences, we contrasted the sequences of the constant portion of the heavy chain of *M. myotis* with human IgG1. As the constant portion of IgG does not assemble from gene segments and lacks multiple genes (as in the case of V genes), we could retrieve the complete sequence from the published genome of one of the species utilized in our study, *M. myotis*. Comparison of the sequences revealed a high degree of

homology between human and bat IgG1 constant portions (approximately 67%). Interestingly, the homology of *M. myotis* with mouse IgG subclasses was lower (58% for IgG2b). These findings explain why Fc-binding proteins specific to human IgG (such as human C1q, human FcRn, and protein G) exhibit also high affinity for bat IgG (as detailed below).

The results from these analyses are now incorporated into the revised version of the manuscript and are depicted in a new figure (Supplementary Figure 10).

- The temperature-dependent binding is a known feature of polyreactive antibodies. Can the authors comment (supported by references or data) on whether bats generally have more polyreactive antibodies compared to humans? The authors mention something in this direction on lines 329 and 330, but it seems that this is a statement based only on differences in the temperature-dependent polyreactivity.

In response to the Reviewer's comment, we conducted additional experiments to assess the frequency of polyreactive antibodies in bat IgG repertoires. In addition to measuring reactivity to dinitrophenol, a well-established molecular probe for detecting polyreactive antibodies, we performed additional ELISA experiments, immunoblot analyses, and immunofluorescence measurements to evaluate the levels of polyreactivity and autoreactivity in the immune repertoires of two bat species, *M. myotis* and *N. noctula*. Together, these assays provided further evidence that the antibody repertoires of the bats exhibit elevated levels of polyreactive antibodies compared to human antibody repertoires.

We have edited the manuscript to incorporate these new data and to clarify issues regarding the polyreactivity of bat IgG. As suggested by the Reviewer, we have included new references supporting our findings. Additionally, new graphs and images have been added to both the main manuscript (Figures 5d, 5e, and 5f) and the supplementary information (Supplementary Figure 13).

- Can you confirm that the gain in polyreactivity at 42 °C is due to binding via the variable antibody domains and not because of an unexpected non-specific interaction via the constant antibody domains of bats?

We appreciate the Reviewer for raising this important question. Indeed, investigating the impact of Fc-dependent interactions of bat IgG was a crucial aspect missing from our manuscript. To address this aspect, we conducted two distinct sets of experiments.

Firstly, we developed an assay to generate F(ab')₂ fragments from bat IgG. This presented a considerable technical challenge as conventional protocols for cleaving human IgG using papain and pepsin were ineffective. We resolved this challenge by digesting bat IgG with the site-specific protease IdeS, successfully producing F(ab')₂ fragments. Immunoassay results revealed that F(ab')₂ fragments from bat IgG exhibited the same temperature sensitivity as intact IgG, strongly suggesting that this phenomenon is Fab region dependent.

Secondly, we investigated the interaction of human and bat IgG with proteins that specifically bind to the constant fragment (Fc) using a surface plasmon resonance-based assay. Previous studies demonstrated that human FcRn can bind to both human IgG and IgG from *M. myotis* and *N. noctula* (PMID: 36272453). Here, we extended this finding to include C1q and protein G. The consistent reactivity of Fc-binding molecules can be attributed to the high sequence homology between bat and human Fc of IgG. These insights enabled us to explore the impact of temperature on Fc-dependent functions of bat IgG. We analyzed the binding of C1q, FcRn, and protein G to human and bat IgG (*M. myotis* and *N. noctula*) as a function of temperature. Our data unequivocally demonstrated that elevated temperature (40 °C) reduced

the binding affinity of all Fc-specific proteins to bat IgGs, mirroring the effect observed with human IgG.

Collectively, the findings from both sets of experiments provide compelling evidence that the phenomenon is Fab-dependent, and the Fc portion is not responsible for altering the antigen-binding activity of bat antibodies at elevated temperatures.

The manuscript has been revised to incorporate these new findings, including a new main figure (Figure 3) and supplementary figure (Suppl. Fig. 10). The results have been described and discussed in the relevant sections of the manuscript.

- The CH2 domains of human IgGs unfold at about 60-70 °C. Can you confirm that the measured melting temperatures are due to thermodynamic stability differences of the Fabs and not the constant antibody domains?

We fully agree with the Reviewer's observation that the measured melting temperatures of bat and human IgG can provide a comprehensive understanding of the unfolding of various fragments and domains. These experiments were not possible due to the significant technical constraint in obtaining enough pure Fab and Fc fragments from bat IgG and due to ethical considerations limiting the sampling of bat serum. Given that our recent findings (refer to responses above) offer supportive evidence for the involvement of the Fab fragment in the temperature-dependent diversification of bat IgG reactivity and considering that interactions through the Fc fragment exhibit similar patterns for bat and human IgG, we believe that separately testing the melting temperatures of Fc and Fab would not yield fundamentally new insights for our current study. Therefore, we would not be able to convince the ethical authorities for collection the needed big amount of blood to perform these experiments.

Nevertheless, we have revised the manuscript to acknowledge that the observed melting temperatures of IgG encompass the entire molecule and caution should be exercised in interpreting differences between human and bat IgG.

- ANS was used to probe the hydrophobicity of proteins, but an orthogonal assay (retention times from hydrophobic interaction chromatography) can support the claim for unusually high hydrophobicity of bat IgGs.

We are thankful to the Reviewer for bringing attention to this aspect of our work. Indeed, using ANS alone may not be the most suitable approach for evaluating the hydrophobicity of bat IgG molecules, especially considering its negatively charged sulfonic group, which could potentially participate in ionic bonds. As recommended by the Reviewer, we employed hydrophobic interaction chromatography to compare the overall hydrophobicity of bat and human IgG. The data revealed no significant difference in the overall hydrophobicity between IgG from humans and bats (specifically *M. myotis*).

Recognizing that this approach assesses only the bulk hydrophobicity of protein molecules, we conducted more precise analyses to evaluate the nature of non-covalent forces involved in the binding of bat IgG to antigens at elevated temperatures. Consequently, we investigated the effect of varying ionic strength on the molecular recognition of three distinct proteins by bat IgG antibodies from *M. myotis* and *N. noctula*. These analyses unveiled a unique pattern of reactivity of bat IgGs as a function of ionic strength, distinct from that observed for human IgG antibodies. This suggests that the interaction of bat antibodies with antigens relies on both polar and hydrophobic forces. Furthermore, the data revealed that at physiological ionic strength, polar interactions are notably reduced, indicating that the formation of complexes with antigens is likely primarily driven by hydrophobic contacts. Notably, this antigen-binding behavior of bat antibodies as a function of ionic strength correlates well with

the behavior of polyreactive antibodies, thereby corroborating our other findings. Interestingly, the implication of hydrophobic interactions also aligns with the sequence analyses of V_H genes of *Myotis* and humans, as described in response to a previous point.

We have revised the manuscript to incorporate and discuss these newly obtained results. The data have been presented in a new figure (Figures 4b, 4c, and 4d).

Minor:

- Some abbreviations are not spelled out at their first mention in the manuscript (e.g., IFN, DAMP). These are standard abbreviations in immunology but not clear for a broad readership.

We have addressed this issue and spelled out the specific and some technical abbreviations in the manuscript.

Reviewer #2 (Remarks to the Author):

The authors' work is very intriguing, which is important in itself for understanding bat immunity, and which might also provide insight into engineering efforts of temperature-sensitive antibodies. But I feel there is a lack of sufficient depth and quality to this work. Here are my specific comments:

We are thankful to the Reviewer for general positive assessment of our work and constructive criticism.

Major comments

(1) The authors' observation in Fig. 1a is certainly very interesting. To corroborate the observations and to make sense of their impact in a broader context, can the authors perform the same study with total IgG from some bird species (flying but not mammals) and from mouse (mammals that don't fly)?

We are grateful to the Reviewer for suggesting the expansion of the number of tested animals to study the sensitivity of their antibodies to temperature. To address this inquiry, we conducted experiments with IgG purified from mouse sera, as well as IgG from three other mammals (rabbit, goat, and cattle). Additionally, we analyzed the thermo-sensitivity of antibodies in the sera of four species of volant birds: *Bubo scandiacus*, *Neophron percnopterus*, *Gyps fulvus*, and *Corvus corone*. The data gathered collectively demonstrated that IgG antibodies from other mammals or birds (in the case of IgY) either did not change or exhibited negligible changes in their antigen-binding reactivity upon temperature variations. Thus, these other animal species displayed behavior identical to that of human IgG. Furthermore, our analyses included IgG purified from two additional bat species, *Myotis blythii* and *Myotis cappaccinii*. Remarkably, temperature significantly increased the reactivity of their antibodies to HEp-2 cells, consistent with the results obtained with IgG from the primary species used in the study, *M. myotis* and *N. noctula*.

We have incorporated the newly generated data into the revised version of the manuscript. The results have been described and discussed in the text and are presented as three new figures (Supplementary Figs. 2, 3, and 7).

(2) Not sure why the authors focus on antibodies. What about T cell receptors of bats? Do they have the same temperature-dependent trait?

We appreciate the Reviewer's suggestion regarding the investigation of another key component of adaptive immunity, the T cell receptor. However, conducting such studies is currently not feasible. The bat species utilized in our study are protected, and we are only authorized to draw small volumes of blood samples. Furthermore, the T cell receptor functions as a membrane receptor, primarily recognizing MHC-peptide complexes displayed on the membranes of other cells. Analyzing the temperature sensitivity in such a complex scenario would necessitate the development of novel tools, such as cell lines, and detection systems that are not yet available for bats.

More broadly, is it possible that all bat proteins (not antibodies, not TCRs) all have this thermodynamic instability? Then all of them are just broken proteins that are sticky at high temperature? Knowing this will help shape more accurate interpretation of the findings of the authors

We concur with the Reviewer that understanding whether the observed phenomenon is specific only to antibodies holds significant relevance. Our newly conducted experiments, aimed at addressing Reviewer 1's query (refer to response to Reviewer 1), revealed that temperature did not exert the same effect on interactions specific to the Fc portion of IgG. For both human IgG and bat IgG, these interactions were reduced at elevated temperatures (40 °C) rather than increased, as observed for Fab-dependent functions. Hence, even within the same molecule, different functional parts do not exhibit identical behavior in response to temperature fluctuations.

To provide further and more direct evidence for the Reviewer's questions, we conducted additional experiments. Specifically, we investigated the interaction of proteins in the cell lysate of bats (*M. myotis*) with autologous albumin (purified from the same bat sera pool as IgG). In contrast to the considerable increase in IgG reactivity at high temperatures, the binding of bat albumin to bat cellular proteins remained unaffected by temperature. This result unequivocally demonstrates that not all proteins from *M. myotis* have the capacity to gain binding at elevated temperatures, thereby ruling out any nonspecific effects of temperature on bat proteins in general.

The newly obtained data have been incorporated into the manuscript and are presented as a new figure (Suppl. Fig. 9).

(3) I think the authors need to dive deep into why the bat antibodies would demonstrate temperature-dependent affinity properties, to give the work sufficient depth. Even just bioinformatics analyses of antibody sequence/structure features would be very helpful, as long as such analyses can provide some novel insight

We are thankful to this Reviewer for the valuable suggestion. A similar idea was also proposed by Reviewer 1. We have addressed this aspect by analyzing the sequences of V_H genes and constant regions of *Myotis* bats, comparing them with those of humans (for full details, please refer to our response to the first question of Reviewer 1).

The newly acquired data strongly support our conclusion regarding the implication of the Fab region in temperature sensitivity. One notable difference observed in the variable region of bats is the degree of hydrophobicity. The involvement of hydrophobic interactions in antigen binding aligns perfectly with the enhanced binding to antigens at high temperatures.

We have referenced literature in the text to demonstrate that the strength of non-polar (hydrophobic) contacts increases with rising temperature.

The results from the sequence analyses have been thoroughly discussed in the revised version of the manuscript and are presented as novel figures (main Figure 4d and Suppl. Figure 10).

(4) The observation with endothelial cells is interesting and looks real. But cells of many other types could also die during inflammation induced by flight. What about opsonization of these cell types? Uncertain why the authors focused on endothelial cells

We agree with Reviewer's suggestion regarding the analysis of other cell types, which is indeed warranted. Our focus on endothelial cells arose from their crucial role in covering blood vessels and their continuous direct contact with circulating antibodies and other blood proteins.

To address this Reviewer's recommendation, we gained access to autologous bat cells for our investigations. To this end, we used the *M. myotis* cell line MmNep. This cell line enabled us to test the reactivity of IgG purified from the same bat species. Additionally, this cell line originates from epithelial cells. Consistent with our experiments using human endothelial cells (HMEC-1), we observed a significant increase in the binding of bat IgG to dying or dead bat cells. This finding further supports our conclusion that bat antibodies could play a role in maintaining tissue homeostasis by preferentially opsonizing dead cells.

We have now incorporated these data into the revised manuscript at appropriate sections. Furthermore, the results have been included as a new figure (Fig. 7) in the main manuscript.

(5) “these data indicate that bat antibodies acquire preferential reactivity to dead 240 endothelial cells at elevated temperatures” What makes the dead cells different from live cells so that bat antibodies would react to them only at high temperature?

Numerous molecular changes have been described to occur during cell death, including alterations in membrane composition, the emergence of neoepitopes, elevation of oxidized lipids on the cell surface, destruction of glycocalyx, and increased membrane permeability, among others. The unique molecular properties of bat antibodies, such as hydrophobicity, flexibility, and polyreactivity, may enable them to recognize one or several of these modifications typical for dead eukaryotic cells. It is noteworthy that natural human IgM antibodies have also been shown to preferentially recognize dead cells (see, for example, PMID: 19058756 and 19363291).

To clarify this important point, we have revised the Discussion section of the manuscript and included appropriate references.

Minor comments

(1) “The experiment 188 demonstrated that bat IgG was characterized by a substantially higher hydrophobicity as 189 compared to human IgG.” Will this be the cause of the temperature-dependent binding properties of bat antibodies? The authors didn't explain the meaning of this finding.

We acknowledge that this portion of our manuscript required clearer articulation, and the conclusions lacked sufficient data support. To address these issues, we conducted more precise analyses to evaluate the nature of non-covalent forces involved in the binding of bat IgG to antigens at elevated temperatures.

Specifically, we investigated the effect of varying ionic strength in the environment on the molecular recognition of three distinct proteins by bat IgG antibodies from *M. myotis* and *N. noctula*. These analyses unveiled a unique pattern of reactivity of bat IgGs as a function of ionic strength, distinct from that observed for human IgG antibodies. This suggests that the interaction of bat antibodies with antigens depends on both polar and hydrophobic forces. Furthermore, the data revealed that at physiological ionic strength, polar interactions are markedly reduced, and the formation of complexes with antigens is primarily driven by hydrophobic interactions.

Importantly, the observed antigen-binding behavior of bat antibodies as a function of ionic strength correlates well with the behavior of polyreactive antibodies, thus providing further support for our findings. Additionally, the implication of hydrophobic interactions aligns with the sequence analyses of V_H genes of *Myotis* and humans, as described in the response to the sequence analyses provided to Reviewer 1.

We have revised the manuscript to describe and discuss these newly obtained results about hydrophobicity in a clearer manner. The data have been presented as new figures (Figures 4b, 4c, and 4d).

REVIEWERS' COMMENTS

Reviewer #1 (Remarks to the Author):

The authors have addressed all my comments. The new data provide more insights that make the manuscript even more important than before.

Reviewer #2 (Remarks to the Author):

I thank the authors for their revisions to address my comments! I have no further comments now, except that Fig. 4b was not mentioned in the main text. The authors need to double check their figure numbering. This is an interesting piece of work!

Tao Wang

Reviewer #1 (Remarks to the Author):

The authors have addressed all my comments. The new data provide more insights that make the manuscript even more important than before.

Reviewer #2 (Remarks to the Author):

I thank the authors for their revisions to address my comments! I have no further comments now, except that Fig. 4b was not mentioned in the main text. The authors need to double check their figure numbering. This is an interesting piece of work!

We are grateful to the reviewers for their positive assessment of our revised manuscript.

As recommended by Reviewer 2 the figure numbering in the manuscript was carefully varified.